# *OptMerge*: Unifying Multimodal LLM Capabilities and Modalities via Model Merging

**Yongxian Wei**[1]    **Runxi Cheng**[1]    **Weike Jin**[2]    **Enneng Yang**[3]    **Li Shen**[3] *

**Lu Hou**[2]    **Sinan Du**[1]    **Chun Yuan**[1]*    **Xiaochun Cao**[3]    **Dacheng Tao**[4]

[1]Tsinghua University; [2]Huawei Noah's Ark Lab;

[3]Sun Yat-sen University; [4]Nanyang Technological University

✉ weiyx23@mails.tsinghua.edu.cn; mathshenli@gmail.com; yuanc@sz.tsinghua.edu.cn

## Abstract

Foundation models update slowly due to resource-intensive training, whereas domain-specific models evolve rapidly between releases. Model merging seeks to combine multiple expert models into a single, more capable model, reducing storage and serving costs while supporting decentralized development. Despite its potential, previous studies have primarily focused on merging visual classification models or Large Language Models (LLMs) for code and math tasks. Recently, Multimodal LLMs (MLLMs) that extend LLMs through large-scale multimodal training have gained traction. However, no benchmark exists for model merging research that clearly divides the tasks of MLLM training and evaluation. In this paper, *(i)* we introduce a model merging **benchmark** for MLLMs, which includes multiple tasks such as VQA, Geometry, Chart, OCR, and Grounding, studying both LoRA and full fine-tuning models. Moreover, we explore how model merging can combine different modalities (*e.g.*, vision-language, audio-language, and video-language models), moving toward the Omni-language model. *(ii)* We implement 10 model merging algorithms on the benchmark. Furthermore, we propose a novel **method** that removes noise from task vectors and robustly optimizes the merged vector based on a loss defined over task vector interactions, achieving an average performance gain of 2.48%. *(iii)* We find that model merging offers a promising way for building improved MLLMs without requiring training data. Our **results** also demonstrate that the complementarity among multiple modalities outperforms individual modalities. All code and checkpoints are publicly available here.

## 1 Introduction

Foundation models experience slow development cycles due to resource-intensive training requirements, while domain-specific models continuously improve during interim periods (Fang et al., 2025). Various developers release their fine-tuned models on open-source communities such as Hugging Face. Model merging (Yadav et al., 2024) aims to combine multiple expert models into a unified model with multiple capabilities. This approach reduces storage and serving costs through model reuse, while supporting decentralized development by enabling independent contributors to build models that can later be merged. Despite its potential, previous studies (Akiba et al., 2025; Ilharco et al., 2023; Yang et al., 2024b) have primarily focused on merging visual classification models across

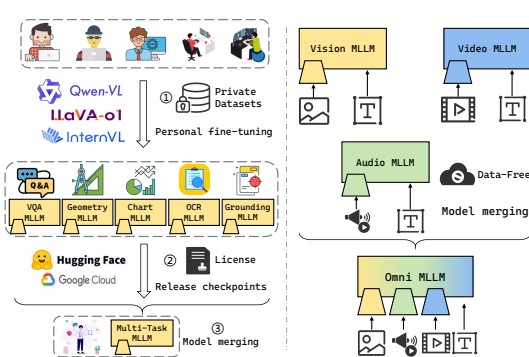

Figure 1: Unifying the **capabilities** or **modalities** of MLLMs from open-source communities via model merging, which is a data-free, cost-effective post-hoc method.

---

*Corresponding author

multiple datasets to extract representations, or merging Large Language Models (LLMs) specifically for code and math tasks.

Recently, Multimodal Large Language Models (MLLMs) that extend LLMs with broader capabilities through large-scale multimodal training have gained traction. Model merging offers a cost-effective way to combine fine-tuned MLLMs with task-specific skills into a unified model (see Fig. 1). However, no benchmark exists for model merging research that clearly divides the tasks of MLLM training and evaluation. Specifically, AdaMMS (Du et al., 2025b) proposes an unsupervised hyper-parameter selection method, but can only merge two MLLMs at a time. For example, merging LLaVA-OneVision-Qwen (Li et al., 2025a) into Qwen2-VL (Wang et al., 2024b) on the Qwen2 architecture. UQ-Merge (Qu et al., 2025) treats each fine-tuning dataset in LLaVA-v1.5 (Liu et al., 2024a) as a separate task without categorization of MLLM capabilities, fine-tuning the base model for each dataset, and using LLaVA-v1.5 as the mixture training baseline.

In this paper, we introduce a model merging benchmark for MLLMs, which includes diverse specialized models such as VQA, Geometry, Chart, OCR, and Grounding. For each corresponding task, we collect comprehensive public datasets with at least 100k samples to ensure effective supervised fine-tuning, and select corresponding benchmarks to evaluate distinct capabilities. We derive an upper bound on the error between the merged model and expert models, proving that merging performance is influenced by the learning rate and iterations, which control the extent of parameter drift. Smaller parameter changes sacrifice task performance but lead to more effortless merging. We choose two types of vision-language models: InternVL2.5 and Qwen2-VL, providing both LoRA and full fine-tuning checkpoints. Moreover, most existing MLLMs specialize in dual modalities, and incorporating new modality encoders requires re-training on new modality-text data. Generating high-quality multimodal instruction data is resource-consuming (Jiang et al., 2024). Therefore, we explore how model merging can efficiently combine different modalities (*e.g.*, vision-language, audio-language, and video-language models), moving toward the Omni-language model. This offers a data-free way to reuse and integrate modality-specific encoders into a unified LLM.

Based on our benchmark, we conduct an in-depth comparison and analysis of state-of-the-art merging methods in capability and modality merging settings. Furthermore, we propose a novel merging method that improves the task vector (*i.e.*, the parameter change between fine-tuned and base models) optimization. *OptMerge* optimizes the merged model based on a loss defined over task vector interactions and applies low-rank approximations to reduce redundant noise, achieving the best results. Combining multiple MLLMs without requiring data, the merged model can even outperform expert MLLMs in their respective capabilities and mixture data training. We also find that merging methods effectively integrate inputs from multiple modalities, outperforming models trained on individual modalities, thus emphasizing the complementary nature of modal information.

In summary, our main contributions are threefold:

- **Benchmark:** We introduce the first model merging benchmark that provides a fine-grained categorization of MLLM capabilities, and evaluates how merging integrates multiple modalities. We train expert models for each task and publicly release their weights and code. This benchmark is designed to help the model merging community better evaluate the generalizability of their methods.
- **Methodology:** We further propose a simple yet effective method, *OptMerge*, which removes noise from task vectors and enhances the robustness of merged vector optimization. Ablation studies show an average performance improvement of 2.48%.
- **Experiments:** We conduct comprehensive experiments and analyses on our benchmark. Our empirical results suggest that model merging can outperform mixture training, offering a viable path to omni-model alignment and a scalable approach to developing MLLMs with reduced computational cost and training time.

## 2 RELATED WORK

**Model merging.** Model merging has emerged as a cost-effective approach to developing improved models by combining multiple expert models to leverage their complementary capabilities (Yadav et al., 2024; Ahmadian et al., 2024). These expert models typically share a common base model, with specialization achieved through fine-tuning on distinct datasets. This approach offers a flexible

and modular method for post-training MLLMs and facilitates the integration of new capabilities into top-performing models. Current research on model merging falls into two primary categories: static merging and dynamic merging. **Static merging** compresses multiple models into a single standard-sized model without adding additional computation or memory overhead. **Dynamic merging** (aka MoE-like methods) (Tang et al., 2024b; Huang et al., 2024; Lu et al., 2024b) requires the dynamic loading of task-specific modules based on test inputs, involving training routers or prior knowledge. The storage parameters for dynamic merging are larger.

Static merging can be further divided into data-free methods and test-time adaptation methods. **Data-free methods** merge fine-tuned models without requiring additional data. We categorize these methods into four groups: (i) Linear interpolation methods that perform arithmetic operations on task vectors (Wortsman et al., 2022; Ilharco et al., 2023; Goddard et al., 2024; Chen et al., 2025); (ii) Sparsification-based methods that reduce redundancy in task vectors (Yadav et al., 2023; Yu et al., 2024; He et al., 2025); (iii) SVD-based methods that identify and exploit the low-rank features of task vectors (Gargiulo et al., 2025; Marczak et al., 2025; Choi et al., 2024; Stoica et al., 2025); and (iv) Optimization-based methods that optimize task vectors via gradient descent (Wei et al., 2025c; Cheng et al., 2025). **Test-time adaptation** (Yang et al., 2024d;c; Daheim et al., 2024) assumes access to unlabeled test datasets, which can be considered a form of transductive learning.

Although test-time adaptation and dynamic merging achieve remarkable results, their practical applicability is limited due to challenges including data privacy concerns, additional storage requirements, and insufficient parallelism in merged models. Therefore, we focus on data-free static merging.

**Model merging in MLLMs.** Recently, several works have attempted model merging for MLLMs, but with different objectives. VL-merging (Sung et al., 2023) merges modality-specific models to create modality-agnostic models, evaluating their effectiveness through fine-tuning on downstream tasks (*e.g.*, image classification). VisionFuse (Chen et al., 2024d) employs task arithmetic to merge LLMs with concatenated visual encoder outputs, primarily focusing on enhancing MLLMs' visual capabilities. Firstly, UnIVAL (Shukor et al., 2023) proposes a study on multimodal model merging via weight interpolation of models trained on different multimodal tasks, showing their benefits in particular for generalization. DAMC (Chen et al., 2024a) composes MLLMs across image, audio, video, and point cloud modalities while reducing modal interference through parameter decoupling.

Several approaches similar to ours aim to merge multiple MLLMs to improve multi-task performance. AdaMMS (Du et al., 2025b) proposes an unsupervised hyperparameter selection method for model merging. However, it requires generating responses for each candidate hyperparameter, making it time-consuming and assuming test set availability during merging. Furthermore, it can only merge two models at a time. For example, merging LLaVA-OneVision-Qwen into Qwen2-VL on the Qwen2 architecture, or merging LLaVA-v1.5 into CogVLM-Chat on the LLaMA architecture. UQ-Merge (Qu et al., 2025) considers uncertainty quantification on text and vision inputs to examine MLLM prediction confidence, requiring unlabeled test sets to calculate prediction and determine merging sequence. This approach measures uncertainty across all candidate models and repeatedly evaluates merged models to find optimal combinations. UQ-Merge treats each fine-tuning dataset in LLaVA-v1.5 (Liu et al., 2024a) as a separate task without categorization of MLLM capabilities, fine-tuning the base model for each dataset and using LLaVA-v1.5 as the mixture training baseline. In contrast, our benchmark collects more comprehensive data with clearer MLLM task divisions for fine-tuning, and we propose a data-free method that requires no hyperparameter search.

## 3 RETHINKING MODEL MERGING

In Sec. 3.1, we begin by introducing common model merging algorithms. In Sec. 3.2, we revisit empirical findings from prior work, and provide a theoretical explanation of the relationship between model fine-tuning and merging performance. Building on this, we analyze the statistical properties of our benchmark, demonstrating both its validity and the challenges it presents.

### 3.1 MERGING BASELINES

Model merging aims to integrate multiple fine-tuned models, all derived from a base model $\boldsymbol{\theta}_0$, into a unified model that consolidates knowledge from diverse sources. Given $n$ fine-tuned models denoted

as $\boldsymbol{\theta}_1, \cdots, \boldsymbol{\theta}_n$, the objective is to produce a single merged model $\boldsymbol{\theta}_m$ that effectively inherits the capabilities of all individual models. We categorize merging methods into four groups.

**Linear interpolation methods: Weight Averaging** (Wortsman et al., 2022) averages the weights of models fine-tuned on different tasks. **Task Arithmetic** (Ilharco et al., 2023) computes task vectors $\boldsymbol{\tau}_i = \boldsymbol{\theta}_i - \boldsymbol{\theta}_0$ for individual tasks and sums them to form a multi-task vector $\boldsymbol{\tau}_m = \sum_{i=1}^{n} \boldsymbol{\tau}_i$. This vector is scaled by a coefficient $\lambda$ and added to the base model $\boldsymbol{\theta}_0$ to obtain the merged model.

**Sparsification-based methods: Ties-Merging** (Yadav et al., 2023) combines steps like trimming, parameter sign determination, and disjoint merging to produce the $\boldsymbol{\tau}_m$. The final model is defined as $\boldsymbol{\theta}_m = \boldsymbol{\theta_0} + \lambda \boldsymbol{\tau}_m$, where $\lambda$ is tuned using the validation set. **DARE** (Yu et al., 2024) randomly drops redundant task vectors and rescales the remaining ones to mitigate parameter interference.

**SVD-based methods: TSV Merging** (Gargiulo et al., 2025) quantifies task-specific feature overlap in weight space by measuring the singular task interference of $\boldsymbol{\tau}_i$. It then reduces task interference through decorrelation. The method seeks orthogonal matrices $V_\perp$ and $U_\perp$ to reconstruct the parameters of the merged model. **Iso-C** (Marczak et al., 2025) proposes an isotropic merging framework that flattens the singular value spectrum of task matrices, and enhances alignment between singular components of task-specific and merged matrices.

**Optimization-based methods: WUDI Merging** Cheng et al. (2025) proves that task vectors $\boldsymbol{\tau}$ form an approximate linear subspace of the fine-tuning data $\boldsymbol{x}$. This property allows the implicit utilization of training data information through task vectors alone. They define layer-wise interference between the merged vector and task vector as $\boldsymbol{\tau}_{m,l} - \boldsymbol{\tau}_{i,l}$ for task $i$ at layer $l$. To optimize the merged vector $\boldsymbol{\tau}_{m,l}$, they minimize this interference $(\boldsymbol{\tau}_{m,l} - \boldsymbol{\tau}_{i,l})\boldsymbol{x}_{i,l}$ with respect to data $\boldsymbol{x}_{i,l}$. Leveraging the linear subspace relationship, they substitute the transpose of $\boldsymbol{\tau}_{i,l}$ for $\boldsymbol{x}_{i,l}$:

$$\min_{\boldsymbol{\tau}_{m,l}} \mathcal{L}_l = \sum_{i=1}^{n} \frac{1}{\|\boldsymbol{\tau}_{i,l}\|_F^2} \left\| (\boldsymbol{\tau}_{m,l} - \boldsymbol{\tau}_{i,l})(\boldsymbol{\tau}_{i,l})^\top \right\|_F^2. \tag{1}$$

This formulates model merging as a data-free optimization problem over parameters. Using the Adam optimizer, we obtain the merged vector $\boldsymbol{\tau}_{m,l}$, which minimizes interference with task vectors on multiple tasks, *i.e.*, the hidden activation satisfies $(\boldsymbol{\theta}_{0,l} + \boldsymbol{\tau}_{m,l})\boldsymbol{x}_{i,l} \approx (\boldsymbol{\theta}_{0,l} + \boldsymbol{\tau}_{i,l})\boldsymbol{x}_{i,l}$.

## 3.2 PARAMETER CHANGES DURING FINE-TUNING MATTER

Model merging exhibits sensitivity to task vectors $\boldsymbol{\tau}_i$ (*i.e.*, parameter changes between fine-tuned models and the base model). Several studies (Yu et al., 2024; Li et al., 2025b) demonstrate that less intensive fine-tuning can yield superior merging performance, even when these models achieve lower accuracy on their respective tasks. In App. B.1, we conduct experiments on the impact of fine-tuning steps on merging performance, and observe that performance tends to rise initially and then decline. This counterintuitive finding suggests that higher-performing expert models do not necessarily produce better merging outcomes. Fine-tuned models tend to converge around the base model in parameter space (Merlin et al., 2023; Chung et al., 2024). When constructing our benchmark, we minimize parameter changes by adjusting the learning rate while maintaining performance improvements on specific tasks. We analyze the upper bound of the loss incurred by model merging:

**Theorem 3.1.** *Consider task $i$ trained for $T$ iterations of gradient descent with a fixed step size $\eta \in (0, 1/L]$, where $L$ is the Lipschitz constant. Let $\gamma := 1 - \eta\mu \in (0, 1)$ denote the PL convergence factor. Then the merged update $\boldsymbol{\tau}_m := \sum_{j=1}^{m} \alpha_j \boldsymbol{\tau}_j$ satisfies*

$$\mathcal{L}_i(\boldsymbol{\Theta} + \boldsymbol{\tau}_m) \leq C_i + \mathcal{O}(\gamma^T) + \mathcal{O}(\delta\,\eta T) + \mathcal{O}(\eta^2 T^2),$$

*where $\mathcal{O}(\gamma^T)$ is the residual error from incomplete convergence on task $i$, $\mathcal{O}(\delta\,\eta T)$ is the cross-task interference term, and $\mathcal{O}(\eta^2 T^2)$ is the curvature term from L-smoothness. This indicates that both the learning rate and iterations influence model merging results. Please refer to App. A for detailed assumptions and proofs.*

*Remark.* Theorem 3.1 provides the first theoretical explanation of how model fine-tuning affects merging performance. The target task's gains dominate in the early training stage, but as convergence approaches, cross-task interference and curvature errors (growing with $\eta T$ and $\eta^2 T^2$) can undermine

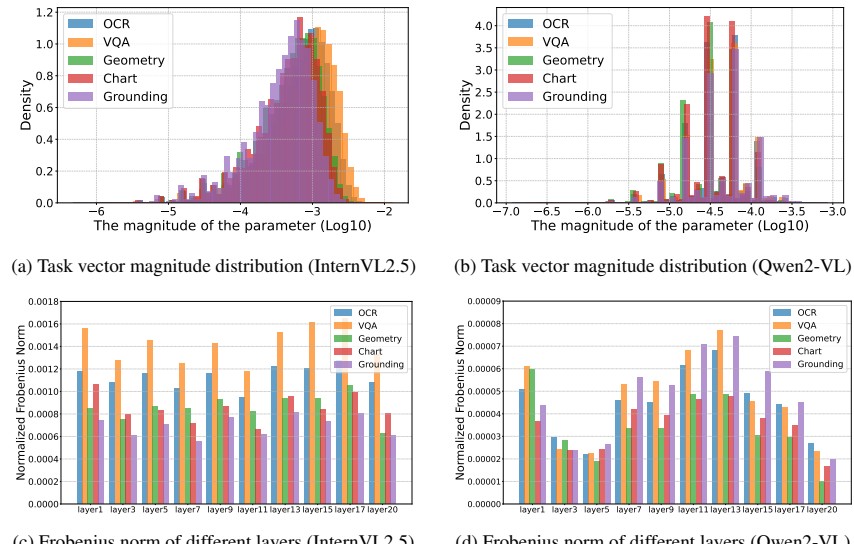

(a) Task vector magnitude distribution (InternVL2.5)  (b) Task vector magnitude distribution (Qwen2-VL)

(c) Frobenius norm of different layers (InternVL2.5)  (d) Frobenius norm of different layers (Qwen2-VL)

Figure 2: **Visualization of task vectors from the benchmark**, revealing the small extent of parameter changes during fine-tuning. InternVL2.5 (full fine-tuning) and Qwen2-VL (low-rank adaptation) exhibit distinct distribution patterns across different tasks.

merging performance. Thus, in the convergence phase, it is essential to control directional leakage (small $\delta$) and limit $\eta T$ to ensure high-quality merging. It supports previous empirical observations: fine-tuned models in current benchmarks typically remain within the same basin near the base model. For example, merging Qwen2.5-Math (Yang et al., 2024a) and Qwen2.5-Coder (Hui et al., 2024) yields poor performance. This is likely due to excessive post-training, which causes significant parameter drift. This insight suggests that poor merging results may not reflect algorithmic flaws, but rather issues with the fine-tuned models. When selecting models from Hugging Face, it's helpful to choose fine-tuned models that are better suited for merging to minimize multi-task degradation.

We examine the weight magnitude distribution of task vectors across our benchmark (see Fig. 2 (a-b)). Our analysis reveals that InternVL2.5, which undergoes full fine-tuning, exhibits a right-skewed distribution. In contrast, Qwen2-VL, fine-tuned using LoRA, displays a multi-modal distribution. This is due to the low-rank nature and scaling factors of LoRA, which constrain the task vectors to be linear combinations in a reduced subspace, causing them to cluster along a few dominant magnitudes. Both models demonstrate distinct magnitude distribution patterns across different tasks. We also compute the normalized Frobenius norm of parameters (*i.e.*, divided by the number of parameters). As shown in Fig. 2 (c-d), the Frobenius norm varies significantly across tasks and layers, which presents challenges that we will address in our approach. The small task vector magnitudes suggest that fine-tuned models and base models exist in adjacent regions of the loss landscape with linear connectivity (Wu et al., 2023), facilitating effective model merging.

## 4 METHODOLOGY

Eq. (1) defines a loss between the merged vector and the task vectors. However, data-free optimization often suffers from instability and convergence issues. To address this, we propose *OptMerge*, a novel method that improves task vector optimization. Specifically, our approach accommodates both full fine-tuning and LoRA fine-tuning scenarios, as they yield model parameters with distinct properties (*e.g.*, low-rank sparsity and varying optimization difficulty). These differences naturally necessitate tailored merging strategies, as detailed in Sec. 4.1 and Sec. 4.2.

### 4.1 MERGING FULL FINE-TUNING MODELS

Task vectors contain significant redundancy and noise, leading to mutual interference during merging. Redundancy stems from different tasks re-learning shared foundational skills, while noise reflects non-essential parameter updates. Directly adding task vectors amplifies these issues, hindering effective

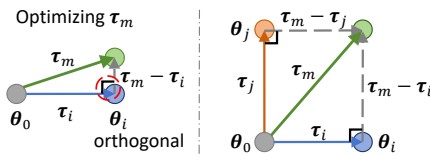

Figure 3: When optimizing Eq. (1), $\boldsymbol{\tau}_m$ tends to take shortcuts by increasing its magnitude to achieve orthogonality.

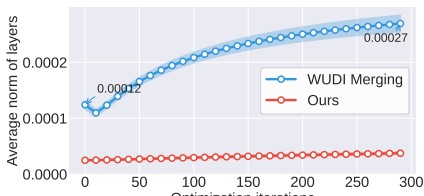

Figure 4: We plot the progression of the Frobenius norm of the merged vector during optimization (average by layers).

merge vector optimization. To address this issue, we propose reducing inter-task interference through low-rank approximation. First, we calculate the average task vector $\bar{\boldsymbol{\tau}}_l = \frac{1}{n}\sum_{i=1}^{n} \boldsymbol{\tau}_{i,l}$ and use it to center task vectors (Choi et al., 2024). Next, we perform SVD to isolate core task-specific knowledge from noise present in the top and lower singular vectors[1].

$$\mathrm{SVD}(\boldsymbol{\tau}_{i,l} - \bar{\boldsymbol{\tau}}_l) = U\Sigma V^\top, \text{ where } U \in \mathbb{R}^{m \times r}, \Sigma \in \mathbb{R}^{r \times r}, V \in \mathbb{R}^{n \times r}. \quad (2)$$

We then apply low-rank approximations to eliminate redundant noise, where $U_{1:k}, \Sigma_{1:k}, V_{1:k}^\top$ represent the top-$k$ singular components. Moreover, we find that substituting $\Sigma_{1:k}V_{1:k}^\top$ for the task vector $\boldsymbol{\tau}_{i,l}$ as the input subspace $\boldsymbol{x}_{i,l}$ allows us to discard secondary row space information, focusing only on the column feature space. Thus, we can optimize $\boldsymbol{\tau}_{m,l}$ via gradient descent on the loss:

$$\min_{\boldsymbol{\tau}_{m,l}} \mathcal{L}_l = \sum_{i=1}^{n} \frac{1}{\|\boldsymbol{\tau}_{i,l}\|_F^2} \left\| (\boldsymbol{\tau}_{m,l} - U_{1:k}\Sigma_{1:k}V_{1:k}^\top - \bar{\boldsymbol{\tau}}_l)(\Sigma_{1:k}V_{1:k}^\top)^\top \right\|_F^2. \quad (3)$$

By truncating singular values, we preserve critical features $V_{1:k}^\top$, which is similar to selecting principal components in Principal Components Analysis (PCA) (Abdi & Williams, 2010). This yields more accurate estimates of $\boldsymbol{x}_{i,l}$ than using $(\boldsymbol{\tau}_{i,l})^\top$.

## 4.2 MERGING LoRA FINE-TUNED MODELS

The inherent low-rank nature of LoRA fine-tuning presents unique optimization challenges for the merge vector. When optimizing $\boldsymbol{\tau}_{m,l}$, gradients become effective only in directions corresponding to non-zero singular values of $\boldsymbol{\tau}_{i,l}$, while approaching zero in other directions (null space). This constraint limits parameter update freedom, preventing $\boldsymbol{\tau}_{m,l}$ from properly exploring the parameter space. We observe that $\boldsymbol{\tau}_{m,l}$ tends to take shortcuts by increasing its magnitude to minimize loss. This occurs because the merge vector must simultaneously accommodate multiple task vectors in different directions. Without constraints, Eq. (1) achieves orthogonality by increasing the length of the merge vector (see Fig. 3). When added to the base model, such large-norm task vectors cause deviation from the original distribution, resulting in collapsed language ability.

To address these challenges, we introduce a set of practical techniques: **(1)** We replace Adam with SGD, which better escapes flat local optima and offers greater stability under sparse gradients. Notably, SGD provides implicit regularization (Smith et al., 2021; Wang et al., 2022), constraining task vector optimization and navigating flat regions induced by null spaces. **(2)** We apply a direct low-rank approximation to $\boldsymbol{\tau}_{i,l}$ using truncated $\mathrm{SVD}(\boldsymbol{\tau}_{i,l}) \approx U_{1:k}\Sigma_{1:k}V_{1:k}^\top$ without centering. The Frobenius norm equals the sum of squared singular values $\|\boldsymbol{\tau}_{i,l}\|_F^2 = \sum_{j=1}^{r} \sigma_j^2$. After truncation, we drop the tail energy $\sum_{j>k} \sigma_j^2$ and thus reduce the norm. **(3)** We also introduce initializing the merged vector with the mean of task vectors to mitigate the issue of excessive merge vector magnitude. As shown in Fig. 4, our approach maintains a relatively consistent norm throughout the optimization process while minimizing loss successfully.

---

[1] We optimize only the model's linear layers, representing each task vector $\boldsymbol{\tau}_{i,l}$ as an $m \times n$ matrix. The remaining layers are merged by simple parameter averaging.

Table 1: **Summary of collected training datasets**, with their corresponding sizes and languages.

| Task Category | Size | Datasets (Language) |
|---|---|---|
| VQA | 588K | GQA (en) (Hudson & Manning, 2019), VQAv2 (en) (Goyal et al., 2017), OKVQA (en) (Marino et al., 2019), LLaVA-Instruct (zh) (Liu et al., 2024a), CogVLM-Singleround (en&zh) (Wang et al., 2024c), CogVLM-Multiround (en&zh) (Wang et al., 2024c) |
| Geometry | 190K | GeoQA+ (zh) (Cao & Xiao, 2022), G-LLaVA (en) (Gao et al., 2023) |
| Chart | 218K | ChartQA (en) (Masry et al., 2022), DVQA (en) (Kafle et al., 2018) |
| OCR | 238K | OCRVQA (en) (Mishra et al., 2019), TextCaps (en) (Sidorov et al., 2020), SynthDoG (en) (Kim et al., 2022), LLaVAR (en) (Zhang et al., 2023), ST-VQA (en) (Biten et al., 2019), TextVQA (en) (Singh et al., 2019), DocVQA (en) (Mathew et al., 2021), DeepForm (en) (Svetlichnaya, 2020), KLC (en) (Stanisławek et al., 2021), TabFact (en) (Chen et al., 2019) |
| Grounding | 135K | RefCOCO (en) (Yu et al., 2016; Mao et al., 2016), VG (en) (Krishna et al., 2017) |

## 5 MLLMs Merging Benchmark

We begin by detailing the benchmark, including its checkpoints, datasets, and evaluation protocols, as well as the implementation of merging algorithms (Sec. 5.1). Next, Sec. 5.2 presents extensive experiments that empirically validate the benchmark, and summarizes the key findings.

### 5.1 Benchmark Details

**Checkpoint construction.** To cover two practical scenarios, namely fine-tuning base models and fine-tuning instruction-tuned models, we select two models that differ in intended use: InternVL2.5-1B-Instruct (Chen et al., 2024c), a lightweight model aligned for instruction following, and Qwen2-VL-7B-Base (Wang et al., 2024b), a general foundation model. Qwen2-VL-7B-Base is among the few publicly available pretrained models, so we use it for the base-model scenario. These choices span different training strategies and scales, enabling a broad assessment of merging methods.

For modality merging, we select Vicuna-7B-v1.5 (Zheng et al., 2023) as the shared LLM. The vision-language model uses CLIP-ViT-L-336px (Radford et al., 2021) as the image encoder, paired with an MLP projection as the connector. The audio-language model adopts BEATs-Iter3+ (Chen et al., 2023) as the audio encoder, with a Q-Former as the connector. The video-language model employs LanguageBind (Zhu et al., 2023) as the video encoder. See App. C for details.

**Training data.** We collect a broader range of domain-specific data, divided into VQA, Geometry, Chart, OCR, and Grounding tasks. The datasets used are summarized in Table 1. For effective supervised fine-tuning, we gather at least 100k public dataset samples for each task, ensuring maximum diversity wherever possible. We process all data into the instruction tuning format.

**Evaluation benchmark.** Current benchmarks (Liu et al., 2024b; Li et al., 2024; Chen et al., 2024b; Fu et al., 2024) predominantly evaluate a model's overall performance but provide limited insights into specific capabilities. Therefore, we curate a carefully selected suite of specialized datasets to evaluate five capabilities: VQA, geometric reasoning, chart understanding, OCR-based VQA, and referring expression grounding. See App. C for details. All evaluation results are obtained using the VLMEvalKit (Duan et al., 2024) and LMMs-Eval (Zhang et al., 2024) libraries under the same settings to ensure fair comparison. All experiments are conducted using $8\times$ NVIDIA V100 GPUs.

For Omni-language models, we assess Audio-VQA, which requires multimodal understanding and spatiotemporal reasoning about visual objects, sounds, and their relationships in videos.

**Merging details.** Following Task Arithmetic (Ilharco et al., 2023), we employ a single coefficient $\lambda$ to scale the merged vector before adding it to the base model. For all model merging methods, we determine the optimal merging coefficient $\lambda$ by searching within the range [0.1, 0.3, 0.5, 0.7, 1.0, 1.5]. In our implementation, the rank size $k$ in Eq. (3) is simply defined as the rank of each task vector divided by the number of tasks (*i.e.*, 5). We use the Adam optimizer with a learning rate of 1e-5 for InternVL while applying the SGD optimizer with a learning rate of 1e-4 for QwenVL. The number of optimization iterations is set to 300. We apply our method exclusively to the linear layer in the model.

### 5.2 Experimental Results

**Capability merging.** As shown in Tables 2 and 3, merging individually specialized models outperforms expert MLLMs on their target tasks. For example, the merged Qwen2-VL achieves 51.05 and

Table 2: **Capability merging results on InternVL2.5 (full fine-tuning) across multiple tasks**. For the merging methods, we highlight the best score in bold and the second-best score in underlining.

| Methods | VQA | | Geometry | | Chart | OCR | | Grounding | | | Avg. |
|---|---|---|---|---|---|---|---|---|---|---|---|
| | VizWiz | GQA (test) | MathVista (mini) | MATH-Vision (mini) | ChartQA (test) | TextVQA (val) | OCRVQA (test) | RefCOCO | RefCOCO+ | RefCOCOg | |
| InternVL2.5-Instruct | 29.15 | 54.62 | 46.80 | 18.42 | 69.48 | 72.51 | 41.08 | 71.69 | 65.41 | 67.40 | 53.66 |
| Individual VQA | 30.58 | 60.91 | 35.50 | 17.11 | 48.76 | 63.68 | 36.04 | - | - | - | 41.80 |
| Individual Geometry | 13.45 | 32.80 | 55.20 | 25.00 | 51.76 | 56.91 | 35.35 | 24.73 | 19.61 | 23.84 | 33.86 |
| Individual Chart | 20.16 | 40.39 | 23.84 | 10.53 | 69.52 | 54.36 | 34.83 | - | - | - | 36.23 |
| Individual OCR | 12.40 | 22.22 | 23.31 | 10.53 | 36.88 | 73.00 | 54.79 | 73.65 | 68.01 | 69.10 | 44.39 |
| Individual Grounding | 19.09 | 25.88 | 28.91 | 14.47 | 41.32 | 58.39 | 74.87 | 76.67 | 71.35 | 70.09 | 48.10 |
| Weight Average | 29.96 | 54.89 | 49.60 | 18.42 | 71.64 | 74.54 | 41.86 | 52.62 | 45.29 | 52.39 | 49.12 |
| Task Arithmetic | 30.67 | 56.34 | 45.36 | 21.05 | 72.88 | 76.26 | 43.39 | 74.90 | 68.15 | 72.75 | 56.18 |
| TIES Merging | 30.63 | 56.48 | 44.50 | 23.68 | 72.28 | 76.29 | 44.01 | 76.01 | 68.45 | 73.65 | 56.70 |
| TA w/ DARE | 30.61 | 56.48 | 48.45 | 21.05 | 73.08 | 76.30 | 43.03 | 74.94 | 68.07 | 73.02 | 56.50 |
| TIES w/ DARE | 30.65 | 56.11 | 43.85 | 27.63 | 72.72 | 76.19 | 43.33 | 75.10 | 68.48 | 73.55 | 56.76 |
| TSV Merging | 31.15 | 56.67 | 52.45 | 28.95 | 70.56 | 75.66 | 45.38 | 65.19 | 58.51 | 59.17 | 54.37 |
| Iso-C | 28.21 | 55.36 | 48.96 | 21.05 | 70.56 | 69.34 | 46.51 | 72.72 | 66.56 | 68.50 | 54.78 |
| WUDI Merging | 31.02 | 56.96 | 53.03 | 17.11 | 69.19 | 75.95 | 46.12 | 76.06 | 70.14 | 74.48 | 57.00 |
| OptMerge (Ours) | 30.97 | 57.13 | 54.48 | 21.05 | 68.72 | 76.01 | 46.35 | 75.97 | 69.72 | 73.94 | 57.44 |
| Mixture Training | 29.79 | 61.33 | 52.83 | 23.68 | 70.32 | 72.96 | 60.25 | 72.06 | 65.93 | 67.46 | 57.66 |

Table 3: **Capability merging results on Qwen2-VL (LoRA fine-tuning) across multiple tasks**. For the merging methods, we highlight the best score in bold and the second-best score in underlining.

| Methods | VQA | | Geometry | | Chart | OCR | | Grounding | | | Avg. |
|---|---|---|---|---|---|---|---|---|---|---|---|
| | VizWiz | GQA (test) | MathVista (mini) | MATH-Vision (mini) | ChartQA (test) | TextVQA (val) | OCRVQA (test) | RefCOCO | RefCOCO+ | RefCOCOg | |
| Qwen2-VL-Base | 5.52 | 5.39 | 47.85 | 23.68 | 0.36 | 20.22 | 1.07 | 45.32 | 37.55 | 31.26 | 21.82 |
| Individual VQA | 41.38 | 62.60 | 33.71 | 28.94 | 66.56 | 80.21 | 55.33 | 39.31 | 32.71 | 38.01 | 47.88 |
| Individual Geometry | 35.57 | 44.63 | 42.50 | 28.95 | 14.56 | 73.95 | 45.96 | 5.57 | 2.31 | 3.90 | 29.79 |
| Individual Chart | 38.58 | 24.24 | 49.28 | 32.89 | 61.08 | 79.75 | 63.67 | 46.28 | 36.67 | 34.06 | 46.65 |
| Individual OCR | 28.38 | 37.53 | 31.81 | 13.16 | 57.40 | 70.50 | 64.68 | 0.59 | 0.46 | 0.26 | 30.48 |
| Individual Grounding | 38.60 | 32.92 | 36.17 | 19.74 | 18.08 | 75.05 | 48.27 | 72.14 | 65.33 | 66.48 | 47.28 |
| Weight Average | 41.47 | 57.33 | 50.21 | 34.21 | 59.56 | 81.09 | 57.85 | 80.72 | 65.37 | 77.68 | 60.55 |
| Task Arithmetic | 40.52 | 62.31 | 40.36 | 26.31 | 79.67 | 81.09 | 59.50 | 75.96 | 61.33 | 75.85 | 60.29 |
| TIES Merging | 41.38 | 59.08 | 46.87 | 34.21 | 67.24 | 81.42 | 58.53 | 80.63 | 65.36 | 77.65 | 61.24 |
| TA w/ DARE | 40.64 | 62.38 | 40.67 | 26.31 | 79.76 | 81.04 | 59.34 | 75.83 | 61.41 | 75.80 | 60.32 |
| TIES w/ DARE | 41.63 | 59.96 | 45.72 | 35.53 | 70.68 | 81.53 | 59.63 | 80.73 | 65.65 | 77.77 | 61.88 |
| TSV Merging | 41.43 | 57.31 | 51.05 | 34.21 | 59.44 | 81.25 | 57.81 | 80.71 | 65.34 | 77.76 | 60.63 |
| Iso-C | 12.31 | 13.44 | 39.96 | 27.63 | 2.80 | 30.05 | 6.12 | 53.68 | 38.96 | 41.90 | 26.69 |
| WUDI Merging | 37.19 | 56.45 | 42.96 | 27.63 | 67.84 | 79.92 | 65.56 | 76.25 | 60.72 | 71.99 | 58.65 |
| OptMerge (Ours) | 41.61 | 61.16 | 48.66 | 40.79 | 74.08 | 81.54 | 60.06 | 80.92 | 65.90 | 78.24 | 63.30 |
| Qwen2-VL-Instruct | 44.09 | 62.18 | 46.02 | 19.73 | 70.04 | 78.38 | 65.42 | 82.89 | 77.87 | 75.63 | 62.23 |

40.79 on Geometry (vs. 42.50 and 28.95 for individual models) and 79.76 on Chart (vs. 61.08). We observe similar gains for OCR and Grounding, with complementary benefits between these tasks. For InternVL2.5-Instruct, we conduct mixture training by combining all task-specific training data for SFT. For Qwen2-VL-Base, we directly use Qwen2-VL-Instruct as the upper bound for mixture training, given its extensive prior SFT with diverse datasets. Notably, our best model merging methods closely match or even surpass mixture training and instruct versions. These results demonstrate that model merging potentially surpasses multi-task learning, while providing a scalable solution for creating high-performing MLLMs with reduced computational cost.

**Categorization of merging methods.** Different merging methods exhibit distinct behaviors. Linear interpolation of task vectors, while ignoring parameter conflicts, is robust but only moderately effective. Sparsification-based methods such as TIES struggle to control sparsity and often underperform relative to task arithmetic. DARE reliably provides plug-and-play gains through simple rescaling. SVD-based methods are sensitive to the spectral structure of task vectors. For example, Iso-C fails on Qwen2-VL because the LoRA-tuned task vectors are already low-rank, and averaging singular values further reduces their Frobenius norm, creating instability in LLMs. Even increasing $\lambda$, as recommended in their paper, only marginally improves results. TSV merging excels in modality merging because its orthogonalization mitigates modal conflicts, but delivers ordinary performance in multi-task settings. In contrast, our approach achieves superior average results across various scenarios, benefiting from stable task vector optimization.

**Improved task vector optimization.** Our method enhances task vector optimization stability, achieving optimal results. In Table 4, we evaluate each component's contribution to overall performance. Starting with WUDI Merging, we incrementally add one component at a time, reporting performance for both LoRA model merging (Qwen2-VL) and modality merging (Vicuna-7B). Replacing Adam with SGD alone does not necessarily improve performance; however, when combined with initializing the merged vector using the mean of task vectors, we observe a significant **4.43%** improvement.

Table 4: **The ablation study.**

| | Qwen2-VL | Vicuna-7B |
|---|---|---|
| WUDI Merging | 58.65 | 64.65 |
| + SGD | 48.88 (-9.77%) | 66.91 (+2.26%) |
| + Initialization | 63.08 (+4.43%) | **67.07** (+2.42%) |
| + Low-rank | **63.30** (+4.65%) | 67.00 (+2.35%) |

Table 5: **Modality merging results on zero-shot image-audio-video question answering tasks** by merging vision-language, audio-language, and video-language models. The "Individual Modalities" columns show baseline performance for each single-modality model.

| Datasets | Individual Modalities | | | Merging Methods | | | | | | | Online Composing | |
|---|---|---|---|---|---|---|---|---|---|---|---|---|
| | Vision | Audio | Video | Weight Average | Task Arithmetic | Ties Merging | TSV Merging | Iso-C | WUDI Merging | OptMerge (Ours) | NaiveMC | DAMC |
| **MUSIC-AVQA** | 50.77 | 27.93 | 49.02 | 47.75 | 52.14 | 50.35 | **53.78** | 52.77 | 52.43 | 53.17 | 53.50 | 52.80 |
| **AVQA** | 75.55 | 47.57 | 79.20 | 69.39 | 78.62 | 75.84 | **80.90** | 77.51 | 76.86 | 80.82 | 80.26 | 80.78 |
| **Avg.** | 63.16 | 37.75 | 64.11 | 58.57 | 65.38 | 63.10 | **67.34** | 65.14 | 64.65 | 67.00 | 66.88 | 66.79 |

Table 6: **Merging results on actual fine-tuned checkpoints collected from Hugging Face**.

| Methods | VQA | | Geometry | | Chart | OCR | | Grounding | | | Avg. |
|---|---|---|---|---|---|---|---|---|---|---|---|
| | VizWiz | GQA (test) | MathVista (mini) | MATH-Vision (mini) | ChartQA (test) | TextVQA (val) | OCRVQA (test) | RefCOCO | RefCOCO+ | RefCOCOg | |
| Qwen2-VL-7B-GRPO-8k | 44.13 | 62.04 | 46.74 | 22.37 | 69.20 | 78.58 | 68.85 | 84.13 | 79.12 | 76.54 | 63.17 |
| Qwen2-VL-7B-Pokemon | 42.51 | 60.96 | 43.69 | 19.74 | 63.20 | 76.75 | 67.64 | 70.11 | 68.80 | 68.64 | 58.20 |
| olmOCR-7B-0225-preview | 43.76 | 61.48 | 38.91 | 18.42 | 67.48 | 77.24 | 68.29 | 75.17 | 71.55 | 69.64 | 59.19 |
| EraX-VL-7B-V1.0 | 36.09 | 54.36 | 38.58 | 25.00 | 56.00 | 70.70 | 65.59 | 41.89 | 40.99 | 43.26 | 47.25 |
| Task Arithmetic | 41.57 | 60.95 | 42.99 | 23.68 | 75.28 | 81.95 | 71.78 | 87.72 | 81.60 | 85.63 | 65.32 |
| TIES Merging | 44.17 | 60.54 | 42.52 | **27.95** | 75.48 | 82.40 | 71.09 | **90.06** | **83.52** | 86.44 | 66.42 |
| TA w/ DARE | 43.33 | 61.15 | 44.37 | 26.95 | 76.48 | 82.93 | **72.00** | 88.93 | 82.79 | 86.07 | 66.50 |
| TIES w/ DARE | **44.37** | 60.78 | 44.37 | 27.63 | 76.04 | 82.61 | 70.93 | 89.40 | 82.93 | **86.77** | 66.58 |
| TSV Merging | 43.73 | **61.40** | 43.54 | 27.94 | 76.44 | 83.15 | 71.65 | 88.53 | 82.25 | 86.41 | 66.50 |
| Iso-C | 43.99 | 61.34 | 40.91 | 22.37 | **76.96** | **83.33** | 71.55 | 87.74 | 82.10 | 85.27 | 65.56 |
| WUDI Merging | 41.39 | 60.11 | 44.20 | 21.05 | 74.36 | 80.78 | 71.12 | 87.96 | 81.50 | 85.48 | 64.80 |
| OptMerge (Ours) | 43.76 | 61.29 | **44.68** | 27.63 | 76.24 | 82.97 | 71.48 | 89.56 | 82.97 | 86.42 | **66.70** |
| Qwen2-VL-Instruct | 44.09 | 62.18 | 46.02 | 19.73 | 70.04 | 78.38 | 65.42 | 82.89 | 77.87 | 75.63 | 62.23 |

Low-rank approximation further enhances performance, demonstrating its effectiveness in preserving critical knowledge from task vectors while maintaining the stability of the Frobenius norm. For full fine-tuned models, Tables 2 and 6 show average improvements of 0.44% and **1.9**% for OptMerge over WUDI Merging, respectively. This highlights the necessity of Eq. (3) over Eq. (1) for denoising task vectors and achieving robust merged-vector optimization.

**Modality merging.** As shown in Table 5, merging methods effectively integrate information from three modalities, outperforming models trained on individual vision, audio, or video inputs. This highlights the complementary nature of modal information and its potential for merging. Online composing dynamically merges activations in the LLM from different modalities during inference, requiring separate parameter storage for each modality (*i.e.*, 3× static merging). NaiveMC (Chen et al., 2024a) performs simple activation averaging, while DAMC (Chen et al., 2024a) decouples parameters during training to reduce modal interference. Notably, the best merging method even outperforms these online composition methods. Advancing Omni models through model merging offers a promising direction for future research.

**Computational requirements.** As illustrated in Table 7, we compare the solving time and GPU memory usage of our approach against mixture training. Our approach optimizes the merged vector over 300 iterations while incurring minimal computational overhead and requiring significantly less GPU memory than data-based training. This effi-

Table 7: **Model merging vs. Data mixing.**

| Methods | Solving Time | GPU Memory |
|---|---|---|
| InternVL2.5-1B (Ours) | 0.22h | 2.62GB |
| InternVL2.5-1B (Mixed) | 25.38h | 240GB |
| Qwen2-VL-7B (Ours) | 3.78h | 21.97GB |
| Qwen2-VL-7B (Mixed) | 24.56h | 256GB |

ciency is achieved through layer-by-layer optimization without requiring training data. Our results confirm that the proposed method is computationally efficient and highly scalable on devices with modern GPUs, facilitating the rapid development of new models based on existing ones.

**Actual checkpoints from Hugging Face.** To evaluate the practicality of model merging in communities, we collect fine-tuned models released by different developers on Hugging Face. Our collection includes a model specialized in math reasoning via multimodal reinforcement learning[2], a personalized model for the Pokemon domain[3], a model focused on converting PDF documents into text[4], and a model with OCR and VQA capabilities in Vietnamese[5]. As shown in Table 6, OptMerge

---

[2]https://huggingface.co/lmms-lab/Qwen2-VL-7B-GRPO-8k

[3]https://huggingface.co/hiyouga/Qwen2-VL-7B-Pokemon

[4]https://huggingface.co/allenai/olmOCR-7B-0225-preview

[5]https://huggingface.co/erax-ai/EraX-VL-7B-V1.0

Table 8: **Ablation study on rank size $k$ ratio for OptMerge**. The rank size $k$ is set to 10%, 20%, 30%, 40%, and 50% of the rank of each task vector.

| $k$ ratio | VQA | | Geometry | | Chart | OCR | | Grounding | | | Avg. |
|---|---|---|---|---|---|---|---|---|---|---|---|
| | VizWiz | GQA (test) | MathVista (mini) | MATH-Vision (mini) | ChartQA (test) | TextVQA (val) | OCRVQA (test) | RefCOCO | RefCOCO+ | RefCOCOg | |
| 10% | 30.90 | 57.26 | 51.49 | 18.42 | 68.40 | 76.10 | 46.39 | 76.36 | 69.99 | 73.96 | 56.93 |
| 20% | 30.97 | 57.13 | 54.48 | 21.05 | 68.72 | 76.01 | 46.35 | 75.97 | 69.72 | 73.94 | 57.43 |
| 30% | 31.55 | 57.15 | 54.50 | 21.05 | 68.72 | 76.27 | 45.67 | 73.63 | 66.84 | 70.92 | 56.63 |
| 40% | 31.49 | 56.92 | 55.77 | 25.00 | 67.36 | 76.06 | 45.96 | 65.55 | 58.40 | 59.64 | 54.22 |
| 50% | 31.37 | 56.68 | 56.75 | 23.68 | 68.08 | 75.81 | 45.02 | 61.45 | 54.80 | 56.19 | 52.98 |

Table 9: **Capability merging results on Qwen2.5-VL-32B-Instruct across multiple tasks**. For the merging methods, we highlight the best score in bold.

| Methods | VQA | | Geometry | | Chart | OCR | | Grounding | | | Avg. |
|---|---|---|---|---|---|---|---|---|---|---|---|
| | VizWiz | GQA (test) | MathVista (mini) | MATH-Vision (mini) | ChartQA (test) | TextVQA (val) | OCRVQA (test) | RefCOCO | RefCOCO+ | RefCOCOg | |
| Qwen2.5-VL-32B-Instruct | 41.39 | 59.34 | 79.21 | **47.36** | 83.64 | 79.62 | 64.58 | 88.01 | 82.41 | 84.06 | 70.96 |
| Individual Geometry | 42.67 | 60.25 | **80.34** | 43.42 | 86.76 | 80.83 | 66.54 | 89.58 | 83.72 | 84.56 | 71.87 |
| Individual Grounding | 41.60 | 59.61 | 78.86 | 42.10 | 85.88 | 79.65 | 64.19 | 88.32 | 82.95 | 84.04 | 70.72 |
| Individual Chart | 43.01 | 61.69 | 74.38 | 43.42 | 86.96 | 81.73 | **67.90** | 89.72 | 83.92 | 84.33 | 71.71 |
| Individual VQA | 42.24 | **62.75** | 78.40 | 42.11 | 86.68 | 81.04 | 67.06 | 89.74 | 83.90 | **84.72** | 71.86 |
| Individual OCR | 42.65 | 61.04 | 75.28 | 34.21 | 87.00 | 81.42 | 67.32 | 89.63 | 83.76 | 84.62 | 70.69 |
| OptMerge (Ours) | **43.52** | 62.50 | 80.01 | 43.42 | **88.92** | **81.91** | 66.37 | **89.94** | **83.97** | 84.68 | **72.52** |

achieves performance that surpasses that of the individual models, effectively integrating knowledge from diverse models to construct a more robust system.

**Rank size $k$.** To further investigate the impact of rank size, we conduct additional ablation studies by setting $k$ to 10%, 20%, 30%, 40%, and 50% of the rank of each task vector. The results are summarized in Table 8. As shown, the performance remains relatively stable for $k$ ratios between 10% and 30%, indicating that OptMerge is robust to moderate changes in rank size.

**Model scales.** We extend our evaluation to the larger Qwen2.5-VL-32B-Instruct model and augment training with additional high-quality fine-tuning data. As shown in Table 9, OptMerge effectively combines multiple fine-tuned models while mitigating cross-task interference, achieving the best overall performance and surpassing the base Qwen2.5-VL-32B-Instruct. These results indicate that OptMerge remains effective and beneficial at larger model scales.

Table 10: Evaluation of the merged model on general multimodal QA benchmarks.

| | MMMU | DocVQA | ScienceQA | AI2D | InfographicVQA |
|---|---|---|---|---|---|
| Individual Geometry | 33.67 | 64.29 | 73.25 | 62.27 | 29.79 |
| Individual Grounding | 34.22 | 65.64 | 76.54 | 63.24 | 33.82 |
| Individual Chart | 30.33 | 57.13 | 40.01 | 29.86 | 26.02 |
| Individual VQA | 26.00 | 62.93 | 50.83 | 44.59 | 39.07 |
| Individual OCR | 38.00 | 77.67 | 63.66 | 54.39 | 41.97 |
| OptMerge (Ours) | **39.33** | **84.18** | **91.89** | **79.44** | **56.84** |

**General tasks.** We further evaluate the merged model (based on InternVL2.5-VL-1B) on a set of general multimodal QA benchmarks that require combinations of multiple abilities. The results are shown in Table 10. On these integrated benchmarks that require multiple abilities, single-ability models cannot solve the tasks effectively. In contrast, our OptMerge, which merges all specialized models, demonstrates emergent integrated capabilities and consistently outperforms the best individual model for each task, with an average improvement of 10.85% across benchmarks.

# 6 CONCLUSION

Model merging aims to combine multiple expert models into a single model without requiring data. In this paper, we introduce the model merging benchmark with detailed categorization of MLLM capabilities, and explore how model merging can effectively combine different modalities of MLLMs. We further propose a novel merging method that effectively removes noise from task vectors and improves the robustness of merged vector optimization. Our results demonstrate that model merging potentially surpasses mixture training, serving as a way for omni-model alignment, while offering a scalable solution for developing MLLMs with reduced computational cost and time.

ACKNOWLEDGMENTS

This work was supported by the National Key R&D Program of China (2022YFB4701400/4701402), SSTIC Grant(KJZD20230923115106012,KJZD20230923114916032,GJHZ20240218113604008). This work is supported by National Key R&D Projects (NO. 2024YFC3307100), NSFC Grant (No. 62576364, No. U2541229), Shenzhen Basic Research Project (Natural Science Foundation) Basic Research Key Project (NO. JCYJ20241202124430041). Dr Tao's research is partially supported by NTU RSR and Start Up Grants.

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

# A   THEORETICAL PROOFS

## NOTATION AND SETTING

**Tasks and losses.**    For tasks $i$, let the loss be $\mathcal{L}_i : \mathbb{R}^d \to \mathbb{R}$ evaluated at parameters $\boldsymbol{\Theta} \in \mathbb{R}^d$.

**Task vectors.**    For task $i$, after $T$ steps of (deterministic) gradient descent (GD) with fixed step size $\eta > 0$ starting from a common initialization $\boldsymbol{\Theta}$, the task vector is

$$\boldsymbol{\tau}_i := -\eta \sum_{t=0}^{T-1} \nabla \mathcal{L}_i(\boldsymbol{\Theta}_t^{(i)}).$$

**Merged update.**    Let the merged vector be

$$\boldsymbol{\tau}_m := \sum_{j=1}^{m} \alpha_j \boldsymbol{\tau}_j,$$

with nonnegative weights $\alpha_j \geq 0$. We study the loss of task $i$ at the merged point $\boldsymbol{\Theta} + \boldsymbol{\tau}_m$.

**Norm and inner product.**    $\|\cdot\|$ is the Euclidean norm and $\langle \cdot, \cdot \rangle$ the Euclidean inner product. For nonzero vectors $u, v$, $\cos(u, v) := \langle u, v \rangle / (\|u\| \|v\|)$.

---

**Assumption A.1** (L-smoothness). Each $\mathcal{L}_i$ has $L$-Lipschitz continuous gradients; that is, for all $\boldsymbol{\Theta}, \boldsymbol{\Theta}'$:
$$\|\nabla \mathcal{L}_i(\boldsymbol{\Theta}) - \nabla \mathcal{L}_i(\boldsymbol{\Theta}')\| \leq L \|\boldsymbol{\Theta} - \boldsymbol{\Theta}'\|.$$
Equivalently, for any $\Delta$:
$$\mathcal{L}_i(\boldsymbol{\Theta} + \Delta) \leq \mathcal{L}_i(\boldsymbol{\Theta}) + \langle \nabla \mathcal{L}_i(\boldsymbol{\Theta}), \Delta \rangle + \frac{L}{2} \|\Delta\|^2.$$

---

**Assumption A.2** (Polyak–Łojasiewicz (PL) condition). Each $\mathcal{L}_i$ satisfies, for some $\mu > 0$:
$$\frac{1}{2} \|\nabla \mathcal{L}_i(\boldsymbol{\Theta})\|^2 \geq \mu \big( \mathcal{L}_i(\boldsymbol{\Theta}) - \mathcal{L}_i^* \big),$$
where $\mathcal{L}_i^* := \inf_{\boldsymbol{\Theta}} \mathcal{L}_i(\boldsymbol{\Theta})$.

---

**Assumption A.3** (Directional similarity). For each $i$:
$$\cos\big( -\nabla \mathcal{L}_i(\boldsymbol{\Theta}), \boldsymbol{\tau}_i \big) \geq \kappa, \quad \kappa \in (0, 1],$$
equivalently:
$$\langle \nabla \mathcal{L}_i(\boldsymbol{\Theta}), \boldsymbol{\tau}_i \rangle \leq -\kappa \|\nabla \mathcal{L}_i(\boldsymbol{\Theta})\| \|\boldsymbol{\tau}_i\|.$$

---

This ensures that the update is indeed a descent direction for task $i$, with alignment quantified by $\kappa$.

---

**Assumption A.4** (Approximate orthogonality). For all $i \neq j$:
$$\cos(\boldsymbol{\tau}_i, \boldsymbol{\tau}_j) \leq \varepsilon, \quad \varepsilon \in [0, 1).$$

---

Prior works (Ilharco et al., 2023; Ortiz-Jimenez et al., 2023) show that task vectors are nearly orthogonal, which helps explain the success of model merging. This likely reflects a general property of high-dimensional spaces: independent directions tend to be almost orthogonal. A small $\varepsilon$ means that tasks are nearly orthogonal in update space, reducing negative transfer effects.

**Assumption A.5** (Bounded gradients). There exists $G > 0$ such that for all $i$ and all $\boldsymbol{\Theta}$ considered:
$$\|\nabla\mathcal{L}_i(\boldsymbol{\Theta})\| \leq G.$$

This assumption is widely adopted in the optimization literature (Gower et al., 2019; Khaled & Richtárik, 2023), where similar boundedness conditions are imposed to control the variance of stochastic gradients and to derive finite-step convergence rates.

**Lemma A.6** (Cross-task cosine leakage). *Under Assumptions A.3–A.4, with $\nabla\mathcal{L}_i(\boldsymbol{\Theta}) \neq \mathbf{0}$ and $\boldsymbol{\tau}_j \neq \mathbf{0}$ to ensure the cosine is well-defined, for $i \neq j$ we have*
$$\big|\cos\big(\nabla\mathcal{L}_i(\boldsymbol{\Theta}), \boldsymbol{\tau}_j\big)\big| \leq \delta, \quad \delta := \kappa\varepsilon + \sqrt{1 - \kappa^2}\,\sqrt{1 - \varepsilon^2}.$$

*Proof sketch.* Normalize $u = -\nabla\mathcal{L}_i/\|\nabla\mathcal{L}_i\|$, $v_i = \boldsymbol{\tau}_i/\|\boldsymbol{\tau}_i\|$, $v_j = \boldsymbol{\tau}_j/\|\boldsymbol{\tau}_j\|$. Use Assumption A.3 to get $\langle u, v_i\rangle \geq \kappa$ and Assumption A.4 to get $\langle v_i, v_j\rangle \leq \varepsilon$, then decompose $u$ and $v_j$ along $v_i$ and its orthogonal complement and apply Cauchy–Schwarz. $\square$

**Lemma A.7** (PL convergence under GD). *Under Assumptions A.1–A.2 and $\eta \in (0, 1/L]$, the GD iterates for task $i$ satisfy*
$$\mathcal{L}_i(\boldsymbol{\Theta}_T) - \mathcal{L}_i^* \leq (1 - \eta\mu)^T\big(\mathcal{L}_i(\boldsymbol{\Theta}_0) - \mathcal{L}_i^*\big).$$

*Proof.* Let one step of GD be $\boldsymbol{\Theta}_{t+1} = \boldsymbol{\Theta}_t - \eta\nabla\mathcal{L}_i(\boldsymbol{\Theta}_t)$. By L-smoothness, for any $x, y$,
$$\mathcal{L}_i(y) \leq \mathcal{L}_i(x) + \langle\nabla\mathcal{L}_i(x), y - x\rangle + \frac{L}{2}\|y - x\|^2.$$

Plug $x = \boldsymbol{\Theta}_t$, $y = \boldsymbol{\Theta}_{t+1}$:
$$\mathcal{L}_i(\boldsymbol{\Theta}_{t+1}) \leq \mathcal{L}_i(\boldsymbol{\Theta}_t) - \eta\left(1 - \frac{L\eta}{2}\right)\|\nabla\mathcal{L}_i(\boldsymbol{\Theta}_t)\|^2.$$

Since $\eta \leq 1/L$, we have $1 - \frac{L\eta}{2} \geq \frac{1}{2}$, thus
$$\mathcal{L}_i(\boldsymbol{\Theta}_{t+1}) \leq \mathcal{L}_i(\boldsymbol{\Theta}_t) - \frac{\eta}{2}\|\nabla\mathcal{L}_i(\boldsymbol{\Theta}_t)\|^2.$$

Apply the PL inequality:
$$\frac{1}{2}\|\nabla\mathcal{L}_i(\boldsymbol{\Theta}_t)\|^2 \geq \mu\big(\mathcal{L}_i(\boldsymbol{\Theta}_t) - \mathcal{L}_i^*\big),$$

to get
$$\mathcal{L}_i(\boldsymbol{\Theta}_{t+1}) - \mathcal{L}_i^* \leq (1 - \eta\mu)\big(\mathcal{L}_i(\boldsymbol{\Theta}_t) - \mathcal{L}_i^*\big).$$
Unrolling the recursion yields the claim. $\square$

**Lemma A.8** (Task vector norm). *If $\boldsymbol{\tau}_j = -\eta\sum_{t=0}^{T-1}\nabla\mathcal{L}_j(\boldsymbol{\Theta}_t^{(j)})$ and $\left\|\nabla\mathcal{L}_j(\boldsymbol{\Theta}_t^{(j)})\right\| \leq G$ for all $t$, then*
$$\|\boldsymbol{\tau}_j\| \leq \eta T G.$$

*Proof.* By the triangle inequality,
$$\|\boldsymbol{\tau}_j\| \leq \eta\sum_{t=0}^{T-1}\left\|\nabla\mathcal{L}_j(\boldsymbol{\Theta}_t^{(j)})\right\| \leq \eta\sum_{t=0}^{T-1}G = \eta T G.$$

$\square$

**Lemma A.9** (Inner-product lower bound). *Under Assumptions A.1–A.2 and $\eta \in (0, 1/L]$,*

$$\langle \nabla \mathcal{L}_i(\boldsymbol{\Theta}), \boldsymbol{\tau}_i \rangle \geq -\left(1 - (1 - \eta\mu)^T\right)\left(\mathcal{L}_i(\boldsymbol{\Theta}) - \mathcal{L}_i^*\right) - \frac{L}{2} \|\boldsymbol{\tau}_i\|^2.$$

*Proof.* Apply the $L$-smooth upper bound with $\Delta = \boldsymbol{\tau}_i$ and rearrange; then use Lemma A.7 to bound $\mathcal{L}_i(\boldsymbol{\Theta} + \boldsymbol{\tau}_i) - \mathcal{L}_i(\boldsymbol{\Theta})$. $\square$

**Theorem A.10** (Finite-step bound). *Consider task $i$ trained for $T$ iterations of gradient descent with a fixed step size $\eta \in (0, 1/L]$. Let $\gamma := 1 - \eta\mu \in (0, 1)$ denote the PL convergence factor. Then the merged update $\boldsymbol{\tau}_m := \sum_{j=1}^m \alpha_j \boldsymbol{\tau}_j$ satisfies*

$$\mathcal{L}_i(\boldsymbol{\Theta} + \boldsymbol{\tau}_m) \leq C_i + \mathcal{O}(\gamma^T) + \mathcal{O}(\delta\,\eta T) + \mathcal{O}(\eta^2 T^2),$$

*where $\mathcal{O}(\gamma^T)$ is the residual error from incomplete convergence on task $i$, $\mathcal{O}(\delta\,\eta T)$ is the cross-task interference term, and $\mathcal{O}(\eta^2 T^2)$ is the curvature term from $L$-smoothness.*

*Proof.* Define the $\eta, T$-independent constant

$$C_i := \mathcal{L}_i(\boldsymbol{\Theta}) - \alpha_i\left(\mathcal{L}_i(\boldsymbol{\Theta}) - \mathcal{L}_i^*\right).$$

By $L$-smoothness,

$$\mathcal{L}_i(\boldsymbol{\Theta} + \boldsymbol{\tau}_m) \leq \mathcal{L}_i(\boldsymbol{\Theta}) + \langle \nabla \mathcal{L}_i(\boldsymbol{\Theta}), \boldsymbol{\tau}_m \rangle + \frac{L}{2} \|\boldsymbol{\tau}_m\|^2.$$

Decomposing the inner product yields

$$\langle \nabla \mathcal{L}_i, \boldsymbol{\tau}_m \rangle = \alpha_i \langle \nabla \mathcal{L}_i, \boldsymbol{\tau}_i \rangle + \sum_{j \neq i} \alpha_j \langle \nabla \mathcal{L}_i, \boldsymbol{\tau}_j \rangle.$$

For the self term, Lemma A.9 implies a constant part absorbed into $C_i$ and a residual term of order $\mathcal{O}(\gamma^T)$, plus a curvature correction $\mathcal{O}(\eta^2 T^2)$ via Lemma A.8. For the cross terms, Lemma A.6 and Assumption A.5 give

$$\left|\langle \nabla \mathcal{L}_i, \boldsymbol{\tau}_j \rangle\right| \leq \delta\,\eta T\, G^2,$$

so the sum over $j \neq i$ is $\mathcal{O}(\delta\,\eta T)$. Finally, $\|\boldsymbol{\tau}_m\| \leq \eta T G \sum_j \alpha_j$ implies the smoothness term is $\mathcal{O}(\eta^2 T^2)$. Combining all contributions yields the stated bound. $\square$

**Theorem A.11** (Bound in the near-convergence regime). *Suppose the residual PL error after $T$ steps is below a given tolerance $\zeta > 0$:*

$$(1 - \eta\mu)^T\left(\mathcal{L}_i(\boldsymbol{\Theta}) - \mathcal{L}_i^*\right) \leq \zeta.$$

*Equivalently,*

$$T \geq \frac{\ln\left((\mathcal{L}_i(\boldsymbol{\Theta}) - \mathcal{L}_i^*)/\zeta\right)}{-\ln(1 - \eta\mu)}.$$

*Then the merged loss satisfies*

$$\mathcal{L}_i(\boldsymbol{\Theta} + \boldsymbol{\tau}_m) \leq C_i + \mathcal{O}(\zeta) + \mathcal{O}(\delta\,\eta T) + \mathcal{O}(\eta^2 T^2),$$

*with the same $C_i$ as in Theorem A.10.*

*Proof.* Starting from Theorem A.10, replace the residual term $\mathcal{O}(\gamma^T)$ by $\mathcal{O}(\zeta)$ using the near-convergence assumption. The cross-task and curvature terms remain unchanged. $\square$

**Remark:** When the learning rate is fixed and the model has not yet converged, the improvement from finetuning on the target task (captured by $1 - \gamma^T$) typically outweighs the influence of other task vectors, especially when these vectors are close to orthogonal (small $\varepsilon$, hence small $\delta$). In this stage, merging different task updates remains stable and can be beneficial.

However, as training approaches convergence, the potential negative impact from other task vectors becomes more significant. Even if each single-task loss continues to decrease, the merged model's loss can worsen due to accumulated cross-task interference, which grows linearly in $T$ as $\mathcal{O}(\delta\,\eta T)$, and curvature effects from $L$-smoothness, which grow quadratically as $\mathcal{O}(\eta^2 T^2)$. This means that over-training on individual tasks can harm the quality of the merged model.

In the convergence regime, a fixed learning rate can lead to increased norms of task vectors. While the single-task losses may remain similar, the larger norms amplify interference and curvature errors, further degrading merged performance. Once the PL error $(1 - \eta\mu)^T(\mathcal{L}_i - \mathcal{L}_i^*)$ falls below a tolerance $\zeta$, the main residual terms are the interference and curvature contributions. Reducing directional leakage (small $\delta$) and controlling the product $\eta T$ are therefore essential for high-quality merging.

# B    MODEL MERGING BENCHMARKS

## B.1    FINE-TUNING INFLUENCE ON MODEL MERGING

To demonstrate the sensitivity of model merging to task vectors $\boldsymbol{\tau}_i$ (*i.e.*, parameter changes between fine-tuned models and the base), we conduct experiments using the standard CLIP-ViT merging benchmark, following the fine-tuning setup of FusionBench (Tang et al., 2024a). We train with Adam (learning rate 1e-5) for 4,000 steps with a batch size of 32. Models are saved every 500 iterations and evaluated for accuracy on the test dataset, as illustrated in Fig. 5. Across eight tasks, convergence typically occurs around 3,000 steps.

Additionally, we evaluate four classical merging methods at various fine-tuning stages, reporting their average accuracy in Fig. 6. The results indicate that increasing the number of fine-tuning steps does not consistently enhance merging performance. Instead, performance typically improves initially before declining. This finding motivates our derivation of Theorem 3.1, which demonstrates that both the learning rate and the number of iterations affect model merging performance. MLLM training is typically organized by full passes over the data (epochs), rather than discrete iteration steps. Accordingly, we set the number of epochs to 1 and reduce the learning rate to limit parameter changes. This keeps the fine-tuned models close to the base model in parameter space while still yielding improvements on specific tasks.

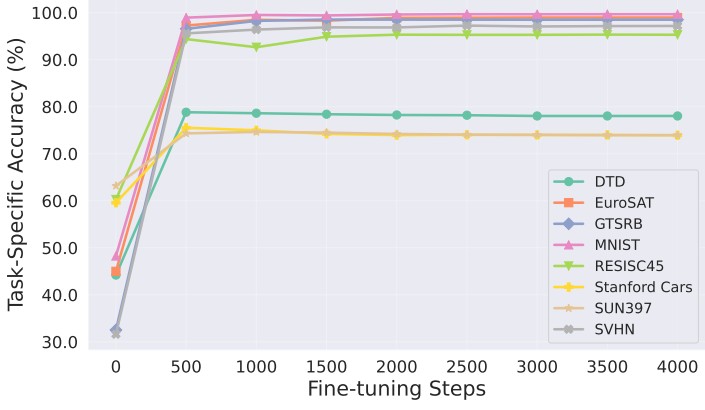

Figure 5: Accuracy of CLIP pre-trained ViT-B/32 fine-tuned separately on eight downstream datasets. As training steps increase, performance on each dataset gradually converges.

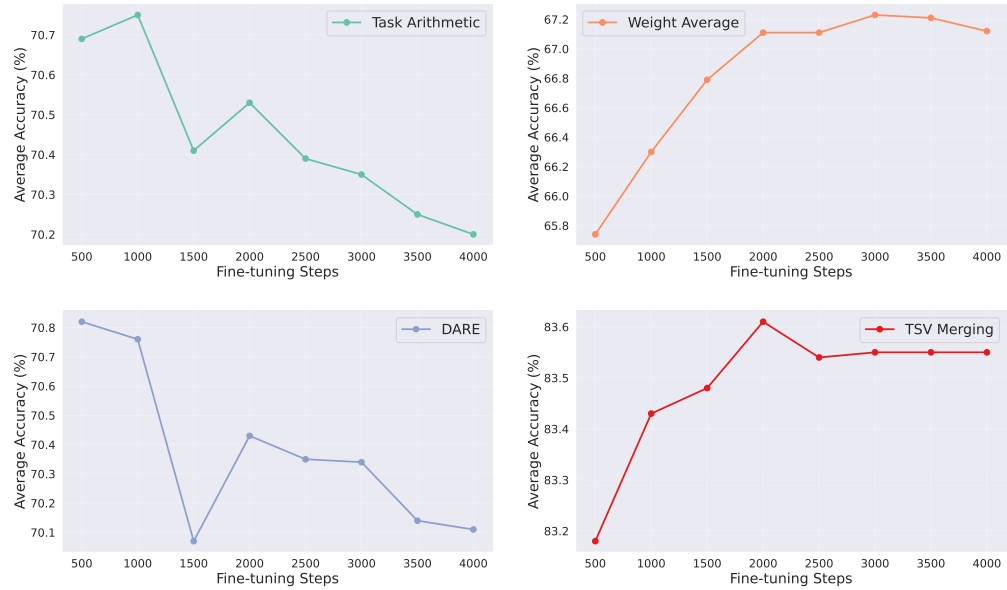

Figure 6: Average accuracy of different model merging methods across eight datasets. Increasing fine-tuning steps does not consistently improve merging performance; instead, performance tends to rise initially and then decline.

## B.2 CHALLENGES OF MLLMS MERGING BENCHMARK

**Disconnection between training and evaluation of MLLMs.** Training and evaluation of MLLMs are developed independently rather than being split from the same dataset. (i) Recent benchmarks (Liu et al., 2024b; Li et al., 2024; Fu et al., 2024) often assess models' comprehensive abilities through pre-defined choice questions, with each benchmark emphasizing different nuanced aspects. (ii) Domain-specific training data is also proprietary and confidential. Consequently, models demonstrate varied capabilities based on their training foundations. For example, LLaVA (Liu et al., 2023) excels in conversational visual reasoning, while InstructBLIP (Dai et al., 2023) performs better on traditional short-answer VQA tasks. These differences present challenges for developing a unified benchmark suitable for multi-task model merging.

**Trade-off between instruction-following and task-specific capabilities.** Public vision datasets provide strong task-specific supervision but rarely use instruction-following formats. Conversely, instruction data generated by models such as GPT-4 (Achiam et al., 2023) often lacks task-specific grounding. This mismatch creates a trade-off between instruction adherence and task expertise.

**Further SFT may lead to overfitting.** Many publicly released models are already instruction-tuned on diverse sources, including open-source, licensed, and private datasets. Additional supervised fine-tuning (SFT) therefore yields diminishing returns and can further overfit models to widely used training distributions (Huang et al., 2025).

## C IMPLEMENTATION DETAILS

**Checkpoint construction.** For InternVL2.5-1B-Instruct, we perform full fine-tuning with a learning rate of 4e-5 and a warmup ratio of 3e-2. For Qwen2-VL-7B-Base, we apply LoRA fine-tuning with a rank of 8, a learning rate of 1e-5, and a warmup ratio of 1e-1. Both models are trained for one epoch using a cosine learning rate scheduler. Different training strategies and scales help evaluate the generalizability of merging methods.

We follow prior work (Chen et al., 2024a) by pairing Vicuna-7B-v1.5 with modality-specific encoders and connectors. Separate models are trained using bi-modal data across three modalities: vision, audio, and video. Additional details are presented in Table 11. The training approach consists of two phases: an alignment stage where only connector parameters are trainable, and a fine-tuning stage where we tune all connector and language model parameters. During fine-tuning, we apply LoRA with a rank of 128 across all linear modules within the LLM. For our merging strategy, we preserve each modality's unique encoder and connector components while merging only the language model parameters, enabling the model to process inputs from all three modalities simultaneously.

Table 11: Overview of modality components and training data.

| Modality | Modality Encoder | Connector | Alignment Data | Fine-tuning Data | Referenced Work |
|---|---|---|---|---|---|
| Vision | CLIP-ViT-L-336px (Radford et al., 2021) | MLP | LCS 558K (Liu et al., 2023) | LLaVA-mixed 665K (Liu et al., 2024a) | LLaVA-1.5 (Liu et al., 2024a) |
| Audio | BEATs-Iter3+ (Chen et al., 2023) | Q-Former | WaveCaps 400K (Mei et al., 2024) | OpenAQA filtered 350K (Gong et al., 2024) | X-InstructBLIP (Panagopoulou et al., 2023) |
| Video | LanguageBind (Zhu et al., 2023) | MLP | LCS 558K (Liu et al., 2023), Valley 702K (Luo et al., 2023) | Video-ChatGPT 100K (Maaz et al., 2024), LLaVA-mixed subset 140K (Liu et al., 2024a) | Video-LLaVA (Lin et al., 2024) |

**Training data.** Following (Chen et al., 2024c), we collect a broader range of domain-specific data, divided into VQA, Geometry, Chart, OCR, and Grounding tasks. For each dataset, we use only the training split, containing question, answer, and image. Samples exceeding 8192 tokens in combined question–answer length or with corrupted images are removed. The remaining data are converted into the ShareGPT instruction-tuning format. Tasks (e.g., VQA, OCR) are trained separately, so cross-task balancing is unnecessary. We collect at least 100k public samples per task to ensure diversity, following common practice for fine-tuning models in the 1B–7B parameter range. Specifically, for grounding tasks, we map coordinates to the [0,1000) range and add the special token notation `<|box_start|><|box_end|>` (Wang et al., 2024b). During Qwen2-VL-Base fine-tuning, we observe that Chinese datasets consistently degraded performance, possibly due to lower data quality or because evaluation benchmarks are primarily in English. Consequently, we use only English datasets for instruction tuning of Qwen2-VL-Base. InternVL2.5-Instruct, already possessing multilingual instruction-following capabilities, is fine-tuned using all available data.

**Evaluation benchmark.** We carefully select specialized datasets to evaluate distinct abilities across tasks. **(i) For VQA**, we utilize VizWiz (Gurari et al., 2018) and GQA (Hudson & Manning, 2019) to assess general visual question answering proficiency. **(ii) For Geometry**, we incorporate multiple challenging subsets: "geometry reasoning", "algebraic reasoning" and "geometry problem solving" from MathVista (Lu et al., 2024a), complemented by "metric geometry - angle", "metric geometry - area", "metric geometry - length" and "solid geometry" from MATH-Vision (Wang et al., 2024a). **(iii) For Chart**, we employ ChartQA (Masry et al., 2022), which tests reasoning and interpretation ability with charts and graphs. **(iv) For OCR**, our evaluation suite includes TextVQA (Singh et al., 2019) and OCRVQA (Mishra et al., 2019). **(v) For Grounding**, we implement referring expression comprehension using RefCOCO (Kazemzadeh et al., 2014), RefCOCO+ (Kazemzadeh et al., 2014), and RefCOCOg (Mao et al., 2016), which require models to identify specific objects in images based on natural language descriptions. All evaluation results are obtained using the VLMEvalKit (Duan et al., 2024) and LMMs-Eval (Zhang et al., 2024) libraries under the same settings to ensure fair comparison. When evaluating MathVista and MATH-Vision benchmarks, we utilize the GPT-4o-mini API to extract answers from the output. The following prompt is used, where {question} denotes the question text and {prediction} denotes the original output from the evaluated model.

```
Please read the following examples. Then extract the answer from the
    model response and type it at the end of the prompt.

Hint: Please answer the question requiring an integer answer and provide
    the final value,
e.g., 1, 2, 3, at the end.
Question: Which number is missing?
```

```
Model response: The number missing in the sequence is 14.
Extracted answer: 14

Hint: Please answer the question requiring a floating-point number with
    one decimal place and provide the final value,
e.g., 1.2, 1.3, 1.4, at the end.
Question: What is the fraction of females facing the camera?
Model response: The fraction of females facing the camera is 0.6,
which means that six out of ten females in the group are facing the
    camera.
Extracted answer: 0.6

Hint: Please answer the question requiring a floating-point number with
    two decimal places and provide the final value,
e.g., 1.23, 1.34, 1.45, at the end.
Question: How much money does Luca need to buy a sour apple candy and a
    butter-scotch candy? (Unit: $)
Model response: Luca needs $1.45 to buy a sour apple candy and a
    butterscotch candy.
Extracted answer: 1.45

Hint: Please answer the question requiring a Python list as an answer and
     provide the final list,
e.g., [1, 2, 3], [1.2, 1.3, 1.4], at the end.
Question: Between which two years does the line graph saw its maximum
    peak?
Model response: The line graph saw its maximum peak between 2007 and
    2008.
Extracted answer: [2007, 2008]

Hint: Please answer the question and provide the correct option letter, e
    .g., A, B, C, D, at the end.
Question: What fraction of the shape is blue?
Choices: (A) 3/11 (B) 8/11 (C) 6/11 (D) 3/5
Model response: The correct answer is (B) 8/11.
Extracted answer: B

{question}
Model response: {prediction}
Extracted answer:
```

For Omni-language models, we select audio-visual question answering task. This task requires multimodal understanding and spatio-temporal reasoning over audio-visual scenes. AVQA (Yang et al., 2022) targets real-world objects and activities. MUSIC-AVQA (Li et al., 2022) specifically focuses on musical performances.

# D DISCUSSIONS

## D.1 UNDERSTANDING THE TASK VECTOR

WUDI Merging (Cheng et al., 2025) substitutes the transpose of the task vector $\tau$ for the input $x$. We reconsider the update process of the task vector, which can be formulated as follows:

$$\tau_{i,l} = \sum_{t=1}^{T} -\eta \cdot \frac{\partial \mathcal{L}(\theta_i^{t-1})}{\partial \theta_{i,l}^{t-1}} \tag{4}$$

$$= \sum_{t=1}^{T} -\eta \sum_{n=1}^{N} \frac{\partial \mathcal{L}(\theta_i^{t-1})}{\partial(\theta_{i,l}^{t-1} x_{i,l}^{n-1})} \cdot \frac{\partial(\theta_{i,l}^{t-1} x_{i,l}^{n-1})}{\partial \theta_{i,l}^{t-1}} \tag{5}$$

$$= \underbrace{\sum_{t=1}^{T} -\eta \sum_{n=1}^{N} \frac{\partial \mathcal{L}(\theta_i^{t-1})}{\partial(\theta_{i,l}^{t-1} x_{i,l}^{n-1})}}_{\text{coefficient}} \cdot (x_{i,l}^{n-1})^{\top}, \tag{6}$$

where $\boldsymbol{\tau}_{i,l}$ denotes the task vector of task $i$ in linear layer $l$, and $\boldsymbol{\theta}_{i,l}^{t-1}$ represents the parameters of task $i$ in linear layer $l$ at time $t-1$. Each parameter in the linear layer can be interpreted as a weighted sum of input vectors across training iterations, with gradients serving as coefficients.

## D.2 LIMITATION AND FUTURE WORK

Due to resource constraints, our experiments were limited to models of 7B parameters. The public datasets we collected may contain lower-quality data. Future work will explore multilingual or reasoning-focused MLLM merging, incorporating visual chain-of-thought datasets (Yang et al., 2025; Li et al., 2025c) to support expert reasoning models. For evaluation, we plan to develop new benchmarks specifically designed to assess the reasoning capabilities of MLLMs.

## D.3 BROADER IMPACTS

Various developers release fine-tuned models on open-source platforms such as Hugging Face (Wolf et al., 2019; Wei et al., 2025b). Model merging reduces storage and serving costs through model reuse and helps preserve data privacy. It also supports decentralized development by enabling independent contributors to train models that can later be merged. We hope this benchmark will help the model merging community better evaluate the generalizability of their methods and accelerate progress in MLLM development.

## E LLM USAGE

This study utilizes large language models to correct grammatical errors.

## F REPRODUCIBILITY STATEMENT

We have open-sourced the code and checkpoints, and provided a detailed description of the implementation details.

