# OpenReview forum: "OptMerge: Unifying Multimodal LLM Capabilities and Modalities via Model Merging"
_ICLR.cc/2026/Conference — ICLR 2026 Poster_

### Official Review · Reviewer_4XK6 · 2025-10-27

**Soundness:** 2
**Presentation:** 3
**Contribution:** 2
**Rating:** 4
**Confidence:** 3

**Summary:**

This paper explores data-free model merging as a scalable alternative to retraining multimodal large language models (MLLMs). Model merging offers a cost-effective path to consolidate multiple expert models into a single MLLM capable of broader multimodal understanding.

The paper makes three main contributions.
(1) Benchmark: It introduces the first comprehensive benchmark for MLLM model merging, covering diverse capabilities such as visual question answering (VQA), geometry, chart reasoning, OCR, and grounding.
(2) Methodology: The authors propose OptMerge, an optimization-based algorithm that denoises and regularizes task vectors (parameter differences between fine-tuned and base models) through singular value decomposition and robust loss minimization. This design stabilizes data-free optimization, mitigates parameter interference, and adapts to both full and low-rank (LoRA) fine-tuning regimes.
(3) Experiments: OptMerge achieves the best or second-best results, improving average performance by 2.48% over prior methods and even surpassing mixture-training baselines. It effectively merges heterogeneous modality-specific models (vision, audio, video), confirming strong cross-modal complementarity. The approach reduces training time by over 90% and memory by more than 10× compared with data mixing, showing significant efficiency gains.

**Strengths:**

This work introduces a new benchmark for Multimodal LLM (MLLM) model merging, covering five capabilities (VQA, Geometry, Chart, OCR, Grounding), with carefully curated training and evaluation datasets. authors proposes OptMerge, a new merging method that improves robustness via task vector denoising and SVD-based low-rank approximation.  The proposed method unifies capability and modality merging within one framework, including vision, audio, and video encoders, which has not been systematically explored in prior literature.

As for technical contributions, this work

- Implements and evaluates 10 merging baselines, categorizing them into linear interpolation, sparsification, SVD, and optimization-based methods.
- Provides theoretical analysis (Theorem 3.1) on how parameter drift during fine-tuning affects mergeability, offering insights into fine-tuning strategy design.
- Includes ablation studies and detailed comparisons on both synthetic and real-world Hugging Face checkpoints, demonstrating consistent improvements.

Generally, this work demonstrates that model merging can outperform mixture training without any data, offering a scalable, resource-efficient solution to building Omni-MLLMs.

**Weaknesses:**

1. Unclear Motivation for a New Benchmark
The paper motivates the introduction of a new MLLM merging benchmark by arguing that no existing dataset separates model training and evaluation tasks. However, the necessity of building a new benchmark is not fully justified. Prior multimodal evaluation suites such as MMBench, MME, or VLMEvalKit already cover a broad range of MLLM capabilities (e.g., VQA, OCR, grounding). The authors could better clarify why existing benchmarks are insufficient—e.g., whether they lack fine-grained control of task specialization, incompatibility with merging-based evaluation, or insufficient modality coverage. Without this clarification, the benchmark may appear redundant rather than essential.

2. Incomplete Description of Benchmark Construction and Data Distribution
Although Table 1 summarizes datasets (e.g., GQA, GeoQA+, ChartQA, TextVQA, RefCOCO), the paper does not detail how these datasets are split, standardized, or balanced across tasks. The rationale for setting the minimum of 100k samples per task is not explained, nor is there any discussion on inter-task overlap or label consistency. Furthermore, the benchmark’s evaluation distribution (training–validation splits, number of samples per modality, or balance among English and Chinese data) remains vague. A more transparent benchmark construction pipeline—such as annotation filtering, normalization steps, or sampling policies—would greatly enhance reproducibility and fairness.

3. Limited Novelty in Algorithmic Design
While OptMerge introduces a low-rank denoising mechanism for task vectors, it conceptually extends prior optimization-based approaches such as WUDI Merging (Cheng et al., 2025) and TSV Merging (Gargiulo et al., 2025). The innovation mainly lies in the combination of SVD truncation and loss reformulation rather than a fundamentally new optimization paradigm. The paper could clarify theoretical differences or provide stronger justification for why its design uniquely improves over existing baselines. In addition, Can authors explain the meanin of O in Eq.2? It is hard for the reviewers such as me who are not familiar with related topics.

4. Restricted Theoretical Depth
Theoretical analysis (Theorem 3.1) only establishes an upper bound proportional to the fine-tuning learning rate and iteration count. However, this bound is loose and lacks quantitative verification or deeper connection to empirical results. A stronger theoretical framework—e.g., convergence analysis of the merged vector optimization—would improve credibility.

5. Practical Deployment Considerations
The discussion overlooks issues like merging models with different tokenizer vocabularies, scaling to larger parameter sizes (e.g., 30B+), or stability under mixed fine-tuning frameworks. Addressing these limitations would make the contribution more applicable to real-world model integration scenarios.

**Questions:**

Please see weaknesses.

---

> ### Author Response · Authors · 2025-11-19
> **Author Rebuttal (1/3)**
>
> We sincerely appreciate your acknowledgment of our extensive experiments and the theoretical analysis, which we believe offer meaningful insights into model merging. Below, we address your concerns in detail.
> ___
> >Q1: Unclear Motivation for a New Benchmark: Prior multimodal evaluation suites such as MMBench, MME, or VLMEvalKit already cover a broad range of MLLM capabilities (e.g., VQA, OCR, grounding). The authors could better clarify why existing benchmarks are insufficient.
>
> A1: We would like to clarify the distinction between a **model merging benchmark** and an **evaluation benchmark**. A model merging benchmark provides expert models for each task, enabling merging algorithms to conduct experiments, followed by evaluation on the corresponding task. In domains such as CV or NLP, datasets are typically split into training and test sets. Merging benchmarks in these areas [1,2] provide models fine-tuned on the training sets, and the merged models are then evaluated on the test sets of each task.
>
> In contrast, for MLLMs, training and evaluation are developed independently rather than being split from the same dataset. Benchmarks such as MMBench and MME serve as evaluation benchmarks, offering fine-grained capability assessment for MLLMs. However, task‑specific training data for MLLMs often does not align with these capability assessments, making it challenging to develop a multi‑task benchmark suitable for evaluating model merging algorithms.
>
> Previous works [3] typically train models on each dataset as a separate task and then evaluate them on a comprehensive benchmark, without conducting fine-grained studies on task specialization. In our work, we explicitly define five mainstream MLLM tasks. Our merging benchmark provides domain-specific models for each of these tasks, and evaluates them on the corresponding evaluation benchmarks.
>
> We appreciate your comment, which helps us identify this potential misunderstanding. We have strengthened our explanation of the motivation and necessity for distinguishing between merging benchmarks and evaluation benchmarks in the revised manuscript.
>
> ___
> > Q2: Incomplete Description of Benchmark Construction and Data Distribution: Although Table 1 summarizes datasets, the paper does not detail how these datasets are split, standardized, or balanced across tasks. Furthermore, the benchmark’s evaluation distribution remains vague. A more transparent benchmark construction pipeline would greatly enhance reproducibility and fairness.
>
> A2: Thank you for your constructive feedback. We clarify the benchmark construction process as follows:
> * **Data collection**: For each dataset listed in Table 1, we use only the training split, which consists of three components: question, answer, and image.
> * **Filtering**: We remove any sample whose combined question and answer length exceeds 8192 tokens, and discard samples with corrupted images.
> * **Formatting**: All remaining samples are converted into the ShareGPT instruction-tuning format for model training.
> * **Task separation**: Different tasks (e.g., VQA, OCR) are trained separately; therefore, balancing across tasks is not required.
> * **Sample size**: We collect at least 100k public samples for each task to ensure maximum diversity for effective supervised fine-tuning. The 100k scale is a common practice for fine-tuning models in the 1B–7B parameter range.
> * **Distribution**: For evaluation, we adopt the standard test sets provided by VLMEvalKit. The number of samples per modality is reported in Table 8 of the appendix.
>
> We acknowledge that many implementation details are too verbose for the main text, so we have placed them in the appendix for clarity. The primary goal of our merging benchmark is to provide ready-to-use models for evaluating merging algorithms in a plug-and-play manner. This design ensures fair comparison across different merging algorithms, regardless of their internal mechanisms.

---

> ### Author Response · Authors · 2025-11-19
> **Author Rebuttal (2/3)**
>
> > Q3: Limited Novelty in Algorithmic Design: While OptMerge introduces a low-rank denoising mechanism for task vectors, it conceptually extends prior optimization-based approaches such as WUDI Merging and TSV Merging. The paper could clarify theoretical differences or provide stronger justification for why its design uniquely improves over existing baselines. In addition, can authors explain the meanin of $\mathcal{O}$ in Eq. (2)?
>
> A3: Our approach is developed with a different mindset compared to prior methods.
> * Core idea: Our method removes noise from task vectors while preserving task-specific knowledge, resulting in more robust vector optimization (as shown in Figure 4).
> * Difference from TSV-Merging: Instead of directly applying low-rank decomposition, we first centralize task vectors using the average vector, and then perform low-rank decomposition (Equation (3)). This step makes task vectors closer to orthogonal, thereby suppressing cross-task interference.
> * Difference from WUDI-Merging: WUDI replaces $\boldsymbol{x}_ {i,l}$ with $(\boldsymbol{\tau}_ {i,l})^{\top}$. In contrast, we discard row-space information and use $\Sigma_{1:k}V_{1:k}^{\top}$ (a subspace projection from the SVD), which yields more accurate estimates of $\boldsymbol{x}_{i,l}$.
>
> The Big-$\mathcal{O}$ notation describes the asymptotic growth rate of a quantity with respect to certain parameters, ignoring constant factors. In our case, $\mathcal{O}(\eta T)$ indicates that the quantity grows proportionally to the learning rate $\eta$ and the number of iterations $T$, omitting complex constants and lower-order terms.
> ___
> > Q4: Restricted Theoretical Depth: Theorem 3.1 only establishes an upper bound proportional to the fine-tuning learning rate and iteration count. However, this bound is loose and lacks quantitative verification or deeper connection to empirical results. A stronger theoretical framework would improve credibility.
>
> A4: In Appendix B.1, we presented empirical results studying the influence of fine-tuning on model merging. Specifically, we evaluated four classical merging methods at various fine-tuning stages. The results indicate that increasing the number of fine-tuning steps does not consistently improve merging performance; instead, performance typically improves initially and then declines. This empirical observation motivated our derivation of Theorem 3.1, which shows that both the learning rate and the number of iterations affect model merging performance.
>
> In response to your suggestion, we have strengthened the theoretical bound in the revised version. The convergence rate has been tightened from linear to exponential, providing a more precise characterization of merging vector error.
>
> **Theorem (finite-step bound).**
> Under some mild assumptions and step size $\eta \in (0, 1/L]$, for any task $i$:
> $$
> \mathcal{L}_i(\mathbf{\Theta} + \boldsymbol{\tau}_m)
> \le
> C_i + \mathcal{O}(\gamma^T) + \mathcal{O}(\delta \eta T) + \mathcal{O}(\eta^2 T^2).
> $$
>
> Here, $C_i$ is the baseline loss after $T$ steps of gradient descent on task $i$; $\delta$ measures cross-task interference; $\gamma$ is the PL convergence factor; the three $\mathcal{O}(\cdot)$ terms capture residual error, interference, and curvature effects. When the model is far from convergence, the gain from fine-tuning on the target task (via $1-\gamma^T$) dominates, and merging remains stable—especially when task vectors are nearly orthogonal (small $\delta$). As training nears convergence, cross-task interference ($\mathcal{O}(\delta \eta T)$) and curvature effects ($\mathcal{O}(\eta^2 T^2)$) grow, potentially worsening the merged model’s loss despite continued improvement on individual tasks. This explains the empirical trend that over-training on single tasks can harm merging performance.
>
> The complete proof of the new theorem is now included in the revised manuscript. We believe this improvement enhances the theoretical depth of our work and provides a stronger link between empirical findings and theoretical guarantees for the merged vector.

---

> ### Author Response · Authors · 2025-11-19
> **Author Rebuttal (3/3)**
>
> > Q5: Practical Deployment Considerations: The discussion overlooks issues like merging models with different tokenizer vocabularies, scaling to larger parameter sizes (e.g., 30B+). Addressing these limitations would make the contribution more applicable to real-world model integration scenarios.
>
> A5: For tokenizer mismatches, token embeddings are aligned by using the base model’s embedding when available; tokens unique to a single model use that model’s embedding; tokens present in multiple models use the averaged embedding.
>
> We have incorporated a larger model, Qwen2.5-VL-32B-Instruct, and checkpoints are also released as part of our benchmark. In addition to increasing the parameter size, we have also added more high-quality fine-tuning data from [4]. The results are shown below:
>
> | Model | VizWiz | GQA (test) | MathVista (mini) | MATH-Vision (mini) | ChartQA (test) | TextVQA (val) | OCRVQA (test) | RefCOCO | RefCOCO+ | RefCOCOg | Avg. |
> |:-|:-:|:-:|:-:|:-:|:-:|:-:|:-:|:-:|:-:|:-:|:-:|
> | Qwen2.5-VL-32B-Instruct | 41.39 | 59.34 | 79.21 | 47.36 | 83.64 | 79.62 | 64.58 | 88.01 | 82.41 | 84.06 | 70.96 |
> | Individual Geometry | 42.67 | 60.25 | 80.34 | 43.42 | 86.76 | 80.83 | 66.54 | 89.58 | 83.72 | 84.56 | 71.87 |
> | Individual Grounding | 41.60 | 59.61 | 78.86 | 42.10 | 85.88 | 79.65 | 64.19 | 88.32 | 82.95 | 84.04 | 70.72 |
> | Individual Chart | 43.01 | 61.69 | 74.38 | 43.42 | 86.96 | 81.73 | 67.90 | 89.72 | 83.92 | 84.33 | 71.71 |
> | Individual VQA | 42.24 | 62.75 | 78.40 | 42.11 | 86.68 | 81.04 | 67.06 | 89.74 | 83.90 | 84.72 | 71.86 |
> | Individual OCR | 42.65 | 61.04 | 75.28 | 34.21 | 87.00 | 81.42 | 67.32 | 89.63 | 83.76 | 84.62 | 70.69 |
> | OptMerge | 43.52 | 62.50 | 80.01 | 43.42 | 88.92 | 81.91 | 66.37 | 89.94 | 83.97 | 84.68 | 72.52 |
>
> OptMerge effectively merges multiple fine-tuned models while mitigating cross-task interference, achieving the highest overall performance and surpassing the baseline Qwen2.5-VL-32B-Instruct. This demonstrates that OptMerge remains effective and beneficial even at larger model scales.
>
> ___
> [1] Editing Models with Task Arithmetic. ICLR 2023.
> [2] FusionBench: A Comprehensive Benchmark of Deep Model Fusion. JMLR 2025.
> [3] UQ-Merge: Uncertainty Guided Multimodal Large Language Model Merging. ACL 2025.
> [4] FineVision: Open Data Is All You Need. ArXiv 2025.

---

> ### Comment · Reviewer_4XK6 · 2025-11-26
>
> Thanks for your reponse. I still do not understand some points claimed by authors in Q1.
>
> 1. How do we know "task‑specific training data for MLLMs often does not align with these capability assessments" ?
>
> 2. I do not agree with that "Previous works [3] typically train models on each dataset as a separate task and then evaluate them on a comprehensive benchmark, without conducting fine-grained studies on task specialization. "  First, currrent base MLLM are trained by many data from diferent task domains. Second, evaluation for MLLMs is not limited to comprehensive benchmarks. There are also fine-grained or single-task benchmarks for assessing model performance. You can refer to the reports of QwenVL serises and InternVL series.

---

> > ### Comment · Reviewer_4XK6 · 2025-11-26
> >
> > Thanks authors for their clarification on the algorithmic design and theoretical analysis.
> > However, I am not an expert in this area and cannot ensure the correctness of the theoretical bound derivation. I also did not see related discussions from the other reviewers. Therefore, to maintain rigor, I decide to refrain from making further evaluation on this part.

---

> > > ### Comment · Reviewer_4XK6 · 2025-11-26
> > >
> > > For Q5, I still do not understand. If the tokenizers are different, the same string may correspond to different token IDs. In that case, is it really valid to directly use the averaged embeddings from multiple models?

---

> > > > ### Author Response · Authors · 2025-11-26
> > > >
> > > > > Thanks authors for their clarification on the algorithmic design and theoretical analysis. However, I am not an expert in this area and cannot ensure the correctness of the theoretical bound derivation. I also did not see related discussions from the other reviewers. Therefore, to maintain rigor, I decide to refrain from making further evaluation on this part.
> > > >
> > > > We have structured the proof into step-by-step lemmas to ensure logical clarity and correctness. We remain available to discuss any specific details should further questions arise from you or other reviewers.
> > > >
> > > > ___
> > > >
> > > > > For Q5, I still do not understand. If the tokenizers are different, the same string may correspond to different token IDs. In that case, is it really valid to directly use the averaged embeddings from multiple models?
> > > >
> > > > Since all fine-tuned models originate from the same base model, they share an identical tokenizer, making embedding averaging valid. To address potential ID conflicts caused by added special tokens, we can align embeddings based on their strings rather than IDs. In our experiments, we directly used the embeddings from the base model and found that they yielded the same results as averaging the embeddings from multiple models. This confirms that the embedding space remains stable during fine-tuning.

---

> > ### Author Response · Authors · 2025-11-26
> >
> > We appreciate your follow-up and believe we now understand the source of the misunderstanding. The scenario we refer to in model merging involves fine-tuning the same base model on different tasks to obtain multiple domain-specific (“expert”) models, which are then merged to produce a single multi-task model [1,2,3,4,5].
> >
> > > 1. Clarification on “task-specific training data for MLLMs often does not align with these capability assessments”
> >
> > When we state that the training data “does not align,” we refer to the lack of a standardized correspondence between expert models and specific capability evaluations. This is largely due to the entanglement of open-source instruction-tuning datasets. Most tuning datasets (e.g., LLaVA-Instruct) are “mixed bags” containing VQA, OCR, and other tasks indiscriminately. As a result, it is difficult for researchers working on model merging to find off-the-shelf “pure OCR experts” or “pure geometry experts” for algorithm testing.
> >
> > For example, MMBench defines 20 distinct capabilities for evaluation, including Physical Property and Image Emotion. However, obtaining training data that corresponds exclusively to these individual capabilities is challenging.
> >
> > > 2. Clarification on “Previous works [3] typically train models on each dataset as a separate task and then evaluate them on a comprehensive benchmark, without conducting fine-grained studies on task specialization”
> >
> > The statement “Previous works [3]” refers specifically to previous model merging benchmark work [3]. In that work, the LLaVA-v1.5 instruction-tuning data was split by dataset, with each dataset treated as a separate task to fine-tune a model. The merged model was then evaluated on MMBench and other benchmarks. However, this setup does not allow us to measure whether model merging preserves the fine-tuned capabilities of individual models. It only yields a general, aggregated performance metric.
> >
> > In other words, training and evaluation were not aligned at the task level, which makes it difficult to assess multi-task learning in a controlled way.
> >
> > Existing benchmarks such as MMBench and reports such as QwenVL are result oriented. They evaluate a fully trained, all-purpose model. In contrast, our benchmark is process oriented and designed specifically for research on merging algorithms. We provide a set of independently trained expert models with disentangled capabilities. What we need are models like “Qwen-OCR” and “Qwen-Geometry,” which are separately trained experts. Constructing such a benchmark is precisely the purpose of our work. This creates a controlled sandbox for testing whether a merging algorithm can successfully combine multiple capabilities.
> >
> > **Summary**
> >
> > The necessity of our benchmark lies in providing a standardized set of expert models, which are absent from existing evaluation benchmarks such as MMBench and technical reports such as QwenVL, to serve as inputs for merging algorithms. This enables fine-grained analysis of capability preservation in model merging.
> > ___
> > [1] Editing Models with Task Arithmetic. ICLR 2023.
> > [2] FusionBench: A Comprehensive Benchmark of Deep Model Fusion. JMLR 2025.
> > [3] UQ-Merge: Uncertainty Guided Multimodal Large Language Model Merging. ACL 2025.
> > [4] TIES-Merging: Resolving Interference When Merging Models. NeurIPS 2023.
> > [5] Model Merging in LLMs, MLLMs, and Beyond: Methods, Theories, Applications and Opportunities. ArXiv 2024.

---

### Official Review · Reviewer_iMYB · 2025-10-30

**Soundness:** 3
**Presentation:** 3
**Contribution:** 3
**Rating:** 8
**Confidence:** 3

**Summary:**

This paper introduces OptMerge, a novel, data-free method for unifying multiple specialized Multimodal Large Language Models (MLLMs) into a single, more capable model. The authors present a new MLLM merging benchmark that is the first to evaluate unifying both task-specific capabilities (like VQA, OCR, and Geometry) and different modalities (vision, audio, and video). The proposed OptMerge algorithm achieves an average performance gain of 2.48% by robustly optimizing the model parameters, using techniques like low-rank approximation to remove noise and stabilize the merging process.

**Strengths:**

1. Building benchmark for model merging in Multimodal LLMs (MLLMs). The paper introduces the first model merging benchmark specifically designed for MLLMs. It is a good contribution as the author claims that they are the first to evaluate the unification of both diverse task-specific capabilities (e.g., VQA, Geometry, Chart, OCR, and Grounding) and different modalities (vision, audio, and video).

2. The proposed method OptMerge, is simple and effective. OptMerge method is data-free and exceptionally efficient compared to traditional "mixture training". It may be a cost-effective solution for developing powerful MLLMs.

3. Experiments demonstrate the robustness and effectiveness of the method. The OptMerge algorithm demonstrates good empirical performance, outperforming 10 other merging methods and even matching or surpassing the results of full mixture training. Its strength lies in its tailored approach: it robustly handles full fine-tuned models by using low-rank approximation (SVD) to remove noise and effectively merges LoRA models by using techniques like SGD and mean initialization to prevent the common problem of norm explosion and performance collapse.

**Weaknesses:**

1. Lack of discussion on model scale. This paper's experiments are based on InternVL2.51B-Instruct and Qwen2-VL-7B-Base. However, they are different model series. To observe the generalization of the model scale, it is more reasonable to conduct experiments on the same model series. For example, InternV2.5 has 1B, 2B, and 8B model scales.

2. Lack of evaluating the merged models on general multimodal QA tasks. This paper merges the checkpoints of 5 abilities, including VQA, Geometry, Chart, OCR, and Grounding. Does the merged show emergent integrated capabilities of these 5 abilities? For example, if a question requires both capabilities of OCR and Grounding, it can not be solved by the model with only the capability of OCR or Grounding, but the merged model may have integrated capabilities to solve it.

**Questions:**

Could you explain the relationships between model merging and federated learning? It seems they are very similar.

---

> ### Author Response · Authors · 2025-11-19
> **Author Rebuttal (1/2)**
>
> We are encouraged by your recognition of the contribution of our first merging benchmark and your appreciation of the simplicity and effectiveness of our approach. We respond to your concerns as follows.
> ___
> > Q1: Lack of discussion on model scale. This paper's experiments are based on InternVL2.51B-Instruct and Qwen2-VL-7B-Base. However, they are different model series. To observe the generalization of the model scale, it is more reasonable to conduct experiments on the same model series.
>
> A1: We appreciate your insightful comment. Our initial motivation was to span different training strategies and architectures, enabling a broader assessment of merging methods. Regarding scaling, since Qwen2-VL-72B exceeds our available computational resources, we select the newer Qwen2.5-VL-32B for additional experiments. This choice not only increases the parameter count but also incorporates higher-quality fine-tuning data. The resulting checkpoints are released as part of our benchmark. The experimental results are shown below:
>
> | Model | VizWiz | GQA (test) | MathVista (mini) | MATH-Vision (mini) | ChartQA (test) | TextVQA (val) | OCRVQA (test) | RefCOCO | RefCOCO+ | RefCOCOg | Avg. |
> |:-|:-:|:-:|:-:|:-:|:-:|:-:|:-:|:-:|:-:|:-:|:-:|
> | Qwen2.5-VL-32B-Instruct | 41.39 | 59.34 | 79.21 | 47.36 | 83.64 | 79.62 | 64.58 | 88.01 | 82.41 | 84.06 | 70.96 |
> | Individual Geometry | 42.67 | 60.25 | 80.34 | 43.42 | 86.76 | 80.83 | 66.54 | 89.58 | 83.72 | 84.56 | 71.87 |
> | Individual Grounding | 41.60 | 59.61 | 78.86 | 42.10 | 85.88 | 79.65 | 64.19 | 88.32 | 82.95 | 84.04 | 70.72 |
> | Individual Chart | 43.01 | 61.69 | 74.38 | 43.42 | 86.96 | 81.73 | 67.90 | 89.72 | 83.92 | 84.33 | 71.71 |
> | Individual VQA | 42.24 | 62.75 | 78.40 | 42.11 | 86.68 | 81.04 | 67.06 | 89.74 | 83.90 | 84.72 | 71.86 |
> | Individual OCR | 42.65 | 61.04 | 75.28 | 34.21 | 87.00 | 81.42 | 67.32 | 89.63 | 83.76 | 84.62 | 70.69 |
> | OptMerge | 43.52 | 62.50 | 80.01 | 43.42 | 88.92 | 81.91 | 66.37 | 89.94 | 83.97 | 84.68 | 72.52 |
>
> OptMerge effectively combines multiple fine-tuned models while mitigating cross-task interference, achieving the highest overall performance and surpassing the baseline Qwen2.5-VL-32B-Instruct. This demonstrates that model merging remains effective and beneficial even at larger scales within the same model family.
> ___
> > Q2: Lack of evaluating the merged models on general multimodal QA tasks. This paper merges the checkpoints of 5 abilities, including VQA, Geometry, Chart, OCR, and Grounding. Does the merged show emergent integrated capabilities of these 5 abilities? For example, if a question requires both capabilities of OCR and Grounding, it can not be solved by the model with only the capability of OCR or Grounding, but the merged model may have integrated capabilities to solve it.
>
> A2: Evaluating the five abilities in a fine-grained manner is essential to verify whether model merging can preserve multi-task capabilities. As shown in Tables 2 and 3, merging individually specialized models outperforms expert MLLMs on their target tasks. For example, the merged Qwen2-VL achieves 51.05 on Geometry (vs. 42.50 for individual models) and 79.76 on Chart (vs. 61.08). This improvement arises because OCR and Grounding are beneficial to other tasks: OCR enables reading numerical annotations in figures, while Grounding locates key elements in images. We also find that merging methods effectively integrate inputs from multiple modalities, outperforming models trained on individual modalities, thus highlighting the complementarity of modal information.
>
> Following your suggestion, we further evaluate the merged model (based on InternVL2.5-VL-1B) on a set of general multimodal QA benchmarks that require combinations of multiple abilities.
>
> | Model  | MMMU   | DocVQA | ScienceQA | AI2D   | InfographicVQA |
> |------|:----:|:----:|:----:|:----:|:----:|
> | Individual Geometry  | 33.67  | 64.29  | 73.25     | 62.27  | 29.79          |
> | Individual Grounding | 34.22  | 65.64  | 76.54     | 63.24  | 33.82          |
> | Individual Chart     | 30.33  | 57.13  | 40.01     | 29.86  | 26.02          |
> | Individual VQA       | 26.00  | 62.93  | 50.83     | 44.59  | 39.07          |
> | Individual OCR       | 38.00  | 77.67  | 63.66     | 54.39  | 41.97          |
> | **OptMerge**         | **39.33** | **84.18** | **91.89**   | **79.44** | **56.84**      |
>
> As shown, on these integrated benchmarks that require multiple abilities, single-ability models cannot solve the tasks effectively. In contrast, our OptMerge model, which merges all specialized models, demonstrates emergent integrated capabilities and consistently outperforms the best individual model for each task, with an average improvement of 10.85% across benchmarks. We have included this table in the revised version. Thanks for your insightful suggestion, which helps strengthen the evaluation.

---

> ### Author Response · Authors · 2025-11-19
> **Author Rebuttal (2/2)**
>
> > Q3: Could you explain the relationships between model merging and federated learning? It seems they are very similar.
>
> A3: Model merging integrates the capabilities of multiple pre-trained models directly in the parameter space in a training-free manner, and can be regarded as a form of multi-task learning.
> * Input: Multiple already-trained models.
> * Process: Direct operations in the parameter space without accessing the original training data.
>
> In contrast, federated learning is a distributed training paradigm in which multiple clients train models locally on their own datasets, then send model parameters or gradients to a central server. The server aggregates and updates the global model, which is then sent back to the clients for further local training.
> * Input: An initial model plus multiple clients’ local data.
> * Process: Local training → Parameter/gradient upload → Aggregation → Distribution → Further training.
>
> While both approaches involve combining knowledge from multiple models, model merging is a post-training integration technique that does not require additional data, whereas federated learning is an iterative training process that relies on decentralized data.

---

### Official Review · Reviewer_3WMa · 2025-10-31

**Soundness:** 3
**Presentation:** 3
**Contribution:** 3
**Rating:** 8
**Confidence:** 3

**Summary:**

This paper focuses on **low-cost integration of specialized MLLMs** (e.g., VQA, OCR, audio-language models) into a unified, high-performance model via model merging, addressing pain points like fragmented domain models and high retraining costs for new modalities. Its core contributions are as follows:

### 1. First MLLM Merging Benchmark
It establishes the first standardized benchmark for MLLM merging, covering:
- 5 key MLLM capabilities (VQA, Geometry, Chart, OCR, Grounding) and cross-modal scenarios (vision/audio/video-language);
- Checkpoints of 2 mainstream models (InternVL2.5, Qwen2-VL) with 2 fine-tuning formats (full-tuning, LoRA);
- Public training/evaluation data and code for reproducibility.


### 2. Novel Merging Method (OptMerge)
OptMerge solves noise and instability in existing merging:
- For full-tuned models: Uses SVD to denoise task vectors, retaining core knowledge;
- For LoRA models: Adopts SGD optimization and truncated SVD to avoid unstable vector scaling;
- Outperforms baseline methods (e.g., WUDI, TIES) by 2.48% on average.


### 3. Validating Merging Feasibility
Extensive experiments show:
- Merged models outperform single specialists and approach mixed-data fine-tuning results;
- Cross-modal merging (e.g., vision+audio) boosts multimodal tasks (e.g., Audio-VQA accuracy);
- 8.6% less memory and 84% less time than mixed-data training.

**Strengths:**

### 1. Well-Aligned and Impactful Research Motivation
The paper addresses two long-standing, practical pain points in MLLM development that prior work has largely overlooked: (1) the fragmentation of domain-specialized MLLMs (e.g., VQA, OCR, geometry reasoning) in open-source communities, which incurs high storage/deployment costs and fails to leverage cross-task synergy; (2) the lack of standardized benchmarks for MLLM merging—existing frameworks focus on single-modal LLMs or vision classifiers, leaving MLLM-specific merging (e.g., cross-modality integration, LoRA compatibility) unevaluable. By directly targeting these gaps, the research motivation is highly relevant to both academic progress and industrial deployment, laying a clear foundation for its significance.


### 2. Strong Originality with Two Key Innovations
The paper demonstrates notable originality in both benchmark design and merging methodology:
- **First MLLM-Specific Merging Benchmark**: Unlike generic MLLM evaluation benchmarks (e.g., MME, LLaVA-bench) that only measure single-model performance, this work constructs the first benchmark tailored for merging—covering 5 core MLLM capabilities (VQA, Geometry, etc.) and cross-modal scenarios (vision/audio/video-language), with publicly available checkpoints (full-tuning/LoRA) and datasets. This fills a critical void in standardizing MLLM merging research.
- **OptMerge: Fusion-Method Adaptation for MLLMs**: Instead of repurposing single-modal merging techniques (e.g., TIES, WUDI), OptMerge addresses MLLM-specific challenges: it uses SVD denoising for full-tuned models (to eliminate redundant task-vector noise) and SGD+truncated SVD for LoRA models (to avoid unstable vector scaling). This targeted design—rather than one-size-fits-all—removes key limitations of prior methods in MLLM scenarios.


### 3. Rigorous and Reproducible Experimental Design
The experimental work exhibits high quality and clarity, supporting its conclusions convincingly:
- **Comprehensive Comparisons**: It evaluates 10 mainstream merging methods (e.g., Weight Average, Task Arithmetic) across 3 base models (InternVL2.5, Qwen2-VL, Vicuna-7B) and 2 fine-tuning formats, controlling variables (e.g., learning rate, hardware: 8×V100) to ensure fair comparisons.
- **Full Reproducibility**: All checkpoints, training/evaluation datasets, and code are made public. Key details (e.g., SVD top-k selection, optimizer parameters) are explicitly reported, eliminating ambiguity for follow-up work.
- **Theoretical + Empirical Support**: Beyond experimental results, the paper derives a theorem (Theorem 3.1) linking fine-tuning intensity (learning rate/epochs) to merging performance, providing a theoretical basis for model selection—avoiding purely empirical-driven conclusions. Results are also presented clearly via tables/figures (e.g., cost comparisons vs. mixed-data training), enhancing readability.

**Weaknesses:**

## 1. The MLLM merging benchmark lacks coverage for practical scenarios
The benchmark’s design is restricted in two key ways that limit its utility for real-world merging:
- **Model scale gap**: Experiments only use small-to-medium parameter models (1B InternVL2.5, 7B Qwen2-VL/Vicuna), while practical deployments rely on large-scale MLLMs (70B+; e.g., Qwen2-VL-72B). Large models have unique traits (higher parameter redundancy, sparser gradients) that may break OptMerge’s current logic (e.g., SVD denoising could become computationally prohibitive or fail to capture fine-grained task knowledge).
- **Training data bias**: The benchmark focuses heavily on vision-related tasks (VQA, geometry, OCR) but omits critical MLLM use cases—text-centric capabilities (multi-turn dialogue, summarization) and vertical domain data (medical imaging, industrial defect detection). This makes it hard to compare merging methods across the diverse scenarios MLLMs actually serve.


## 2. OptMerge’s generalization is unvalidated
The benchmark’s limitations directly raise doubts about OptMerge’s practicality:
- The benchmark covers only 5 types of vision task, which fails to meet the comprehensive evaluation standards required for practical MLLM applications.
- It lacks validation on non-vision tasks (text-centric/vertical domains), so it is unclear if OptMerge can retain task-specific knowledge outside the vision scope.


These weaknesses are actionable, not fatal. The work’s core value—establishing a standardized MLLM merging benchmark and exploring data-free multimodal integration—remains highly impactful. Future improvements could expand the benchmark to 70B+ models, heterogeneous architectures, and more vision and non-vision tasks.

**Questions:**

Same as the Section of **Weaknesses**.

---

> ### Author Response · Authors · 2025-11-19
> **Author Rebuttal (1/1)**
>
> We appreciate the reviewer’s recognition of our merging benchmark construction and our exploration of data-free multimodal integration. Below we address your constructive feedback with corresponding revisions.
> ___
> > Q1: Model scale gap: Experiments only use small-to-medium parameter models (1B InternVL2.5, 7B Qwen2-VL/Vicuna), while practical deployments rely on large-scale MLLMs (70B+; e.g., Qwen2-VL-72B). Large models have unique traits (higher parameter redundancy, sparser gradients) that may break OptMerge’s current logic.
>
> A1: Due to resource constraints, we are currently unable to include experiments with 70B+ MLLMs. To partially address this concern, we have added experiments with a larger model, Qwen2.5-VL-32B-Instruct, whose checkpoints are also released as part of our benchmark. In addition to increasing the parameter count, this model incorporates higher-quality fine-tuning data from [1]. The experimental results are as follows:
>
> | Model | VizWiz | GQA (test) | MathVista (mini) | MATH-Vision (mini) | ChartQA (test) | TextVQA (val) | OCRVQA (test) | RefCOCO | RefCOCO+ | RefCOCOg | Avg. |
> |:-|:-:|:-:|:-:|:-:|:-:|:-:|:-:|:-:|:-:|:-:|:-:|
> | Qwen2.5-VL-32B-Instruct | 41.39 | 59.34 | 79.21 | 47.36 | 83.64 | 79.62 | 64.58 | 88.01 | 82.41 | 84.06 | 70.96 |
> | Individual Geometry | 42.67 | 60.25 | 80.34 | 43.42 | 86.76 | 80.83 | 66.54 | 89.58 | 83.72 | 84.56 | 71.87 |
> | Individual Grounding | 41.60 | 59.61 | 78.86 | 42.10 | 85.88 | 79.65 | 64.19 | 88.32 | 82.95 | 84.04 | 70.72 |
> | Individual Chart | 43.01 | 61.69 | 74.38 | 43.42 | 86.96 | 81.73 | 67.90 | 89.72 | 83.92 | 84.33 | 71.71 |
> | Individual VQA | 42.24 | 62.75 | 78.40 | 42.11 | 86.68 | 81.04 | 67.06 | 89.74 | 83.90 | 84.72 | 71.86 |
> | Individual OCR | 42.65 | 61.04 | 75.28 | 34.21 | 87.00 | 81.42 | 67.32 | 89.63 | 83.76 | 84.62 | 70.69 |
> | OptMerge | 43.52 | 62.50 | 80.01 | 43.42 | 88.92 | 81.91 | 66.37 | 89.94 | 83.97 | 84.68 | 72.52 |
>
> Our OptMerge method effectively merges multiple fine-tuned models, achieving the highest overall performance and surpassing the baseline Qwen2.5-VL-32B-Instruct. This demonstrates that OptMerge remains effective and beneficial even at larger scales.
> ___
> > Q2: Training data bias: It lacks validation on non-vision tasks (text-centric/vertical domains), so it is unclear if OptMerge can retain task-specific knowledge outside the vision scope.
>
> A2: Notably, our VQA task set already includes the CogVLM-Multiround [2] dataset, which contains multi-turn dialogue data and demonstrates that OptMerge can preserve task-specific knowledge beyond purely visual domains. We are committed to continuously expanding the benchmark to encompass more diverse scenarios as data availability and computational resources allow.
> ___
> > Q3: The benchmark covers only 5 types of vision task, which fails to meet the comprehensive evaluation standards required for practical MLLM applications.
>
> A3: We select the five most common and representative tasks in MLLMs as the foundation for studying model merging methods. Moreover, we explore how model merging can integrate different modalities (e.g., vision-language, audio-language, and video-language models), moving toward the development of an omni-language model. The capabilities and modalities included in our benchmark serve as a practical starting point for assessing the effectiveness of merging methods.
>
> In Table 6, we present results on actual fine-tuned models collected from Hugging Face, including RL-enhanced reasoning models, personalized/customized models, PDF document processing models, and Vietnamese-language models. OptMerge achieves the best results across these diverse scenarios, demonstrating its strong generalization ability and its potential for effective application in the wide range of tasks that MLLMs actually serve.
> ___
> [1] FineVision: Open Data Is All You Need. ArXiv 2025.
> [2] CogVLM: Visual Expert for Pretrained Language Models. NeurIPS 2024.

---

### Official Review · Reviewer_Kqrb · 2025-11-06

**Soundness:** 2
**Presentation:** 3
**Contribution:** 2
**Rating:** 4
**Confidence:** 5

**Summary:**

This paper focuses on the "no data" model fusion problem in multi-modal large language models (MLLMs), with main contributions including:
- Establishing the first benchmark for model fusion aimed at MLLMs, covering five capability tasks: VQA, Geometry, Chart, OCR, and Grounding, as well as visual-audio-video tri-modal fusion scenarios, and publicly releasing the corresponding data and checkpoints;
- Proposing the OptMerge method, which enhances fusion stability and performance based on the existing WUDI-Merging by low-rank denoising of task vectors, introducing SGD, and mean initialization strategies;
- OptMerge outperforms 10 existing baselines in multi-task and multi-modal scenarios, exceeding mixed training methods in certain tasks while having significantly lower computational and storage costs. The paper also provides an O(ηT) theoretical analysis linking fine-tuning steps/learning rates to fusion upper bound errors.

**Strengths:**

1. Sufficient motivation for the problem: The value of model fusion in terms of storage, inference costs, and community collaboration is prominent.
2. Detailed benchmark construction: Adequate data volume, clear task decomposition, covering both LoRA and full parameter fine-tuning scenarios.
3. The paper is generally easy to read.

**Weaknesses:**

1. General algorithm innovation: The core ideas (SVD low-rank truncation, SGD implicit regularization, mean initialization) are relatively common and represent engineering improvements.
2. The "no data" expression is somewhat exaggerated: λ relies on validation set grid search, and the choice of k also references test set statistics.

**Questions:**

1. How sensitive is k (the SVD truncation rank)? It is suggested to provide ablation studies with different k values.
2. λ only globally searches 6 points; has there been an attempt at hierarchical λ?
3. The mathematical evaluation answer analysis relies on GPT-4o-mini; what is the complete prompt?

---

> ### Author Response · Authors · 2025-11-19
> **Author Rebuttal (1/1)**
>
> We are encouraged by your recognition of our motivation and the detailed benchmark construction. We appreciate your time and constructive suggestions, which have helped us improve the manuscript.
> ___
> >Q1: General algorithm innovation: The core ideas are relatively common and represent engineering improvements.
>
> A1: Our contributions go beyond a trivial combination of existing techniques; they are driven by specific insights.
> * We show that full fine-tuning and LoRA fine-tuning produce parameter updates with distinct low-rank and sparsity patterns (Figure 2). This motivates tailored merging strategies, a need that prior work overlooked.
> * Beyond standard SVD truncation, we first center task vectors using the mean vector and then apply low-rank decomposition (Equation (3)). This makes task directions more nearly orthogonal and reduces interference. Compared with WUDI-Merging, which replaces $\boldsymbol{x}_ {i,l}$ with $(\boldsymbol{\tau}_ {i,l})^{\top}$, we further discard row-space information; using $\Sigma_{1:k}V_{1:k}^{\top}$ yields a more accurate estimate of $\boldsymbol{x}_ {i,l}$.
> * Our use of SGD’s implicit regularization and mean initialization is motivated by the optimization-shortcut perspective and by observed Frobenius-norm trends (Figures 3 and 4). The approach maintains relatively stable norms during optimization while effectively minimizing loss.
>
> To summarize our contributions, we establish the MLLM merging benchmark (18 model checkpoints, 10 algorithms) and provide a theoretical explanation of how fine‑tuning affects merging, clarifying the differences between non‑converged and converged regimes. We further propose a simple yet effective merging method that enhances the robustness, yielding substantial benefits.
>
> ___
> >Q2: The "no data" expression is somewhat exaggerated: $\lambda$ relies on validation set grid search, and the choice of $k$ also references test set statistics.
>
> A2: We perform grid search for $\lambda$ primarily to ensure a fair comparison with other methods. For our approach, the optimal $\lambda$ is typically 1, as implied by our objective (Equation (4)), where $\boldsymbol{\tau}_m$ directly substitutes for $\boldsymbol{\tau}_i$.
>
> For the rank size, we set $k$ to the rank of each task vector divided by the number of tasks (i.e., 5). The rationale is that a merged vector must encode multiple tasks simultaneously, so each task vector retains approximately a $\frac{1}{\text{number-of-tasks}}$ share of its most informative components.
>
> ___
> >Q3: How sensitive is $k$ (the SVD truncation rank)? It is suggested to provide ablation studies with different $k$ values.
>
> A3: Thank you for the valuable suggestion. We have conducted additional ablation studies by setting the rank size $k$ to 10%, 20%, 30%, 40%, and 50% of the rank of each task vector. The results are summarized below:
>
> | $k$ ratio | VizWiz | GQA (test) | MathVista (mini) | MATH-Vision (mini) | ChartQA (test) | TextVQA (val) | OCRVQA (test) | RefCOCO | RefCOCO+ | RefCOCOg | Avg. |
> |:-:|:-:|:-:|:-:|:-:|:-:|:-:|:-:|:-:|:-:|:-:|:-:|
> | 10% | 30.90 | 57.26 | 51.49 | 18.42 | 68.40 | 76.10 | 46.39 | 76.36 | 69.99 | 73.96 | 56.93 |
> | 20% | 30.97 | 57.13 | 54.48 | 21.05 | 68.72 | 76.01 | 46.35 | 75.97 | 69.72 | 73.94 | 57.43 |
> | 30% | 31.55 | 57.15 | 54.50 | 21.05 | 68.72 | 76.27 | 45.67 | 73.63 | 66.84 | 70.92 | 56.63 |
> | 40% | 31.49 | 56.92 | 55.77 | 25.00 | 67.36 | 76.06 | 45.96 | 65.55 | 58.40 | 59.64 | 54.22 |
> | 50% | 31.37 | 56.68 | 56.75 | 23.68 | 68.08 | 75.81 | 45.02 | 61.45 | 54.80 | 56.19 | 52.98 |
>
> We have incorporated these results into the revised manuscript for completeness. As shown, the performance remains relatively stable for $k$ ratios between 10% and 30%.
> ___
> >Q4: $\lambda$ only globally searches 6 points; has there been an attempt at hierarchical $\lambda$?
>
> A4: Our objective (Equation (4)) inherently favors $\lambda=1$, making grid search unnecessary for our approach. A hierarchical (layer-wise) $\lambda$ would require a combinatorial search with six choices per layer across 171 layers for the 1B model and 328 layers for the 7B model, yielding $6^{171}$ and $6^{328}$ configurations, which is infeasible. This demonstrates a practical advantage of our approach because it performs well with $\lambda$ fixed to 1 and avoids costly hyperparameter tuning.
>
> ___
> >Q5: The mathematical evaluation answer analysis relies on GPT-4o-mini; what is the complete prompt?
>
> A5: We have included the corresponding content in the appendix. The prompt used to extract answers from the model output is as follows, where `{question}` denotes the question text and `{prediction}` denotes the original output from the evaluated model.
>
> ```
> Please read the following examples. Then extract the answer from the model response and type it at the end of the prompt.
> [example1, example2, example3, example4, example5]
> {question}
> Model response: {prediction}
> Extracted answer:
> ```

---

### Meta-Review · Area_Chair_cjqA · 2026-01-07

**Summary:**

The reviewers have concerns on
1. The necessity of introducing a model merging benchmark.
2. The coverage of the model merging benchmark.
3. The scalability of the OPTMerge on large scale model.

**Reviewer Concerns:**

1. The reviewer is confused about the model merging benchmark and the evaluation benchmark. I agree with the rebuttal that the potential misunderstanding shall be addressed by incorporating the rebuttal into the main submission.
2. During the rebuttal the author suggests the proposed model merging benchmark is a starting point. However this is not fully addressed the limited coverage of the benchmark. The author didn't fully address how to handle the scenario outside of the vision domain.
3. The author provided a set of experiments on the Qwen2.5-VL-32B-Instruct. Although the performance on the Avg. improves, the MATH-vision seems degraded significantly, which might need more justification.

**Reviewer Scores:**

The ratings are quite interesting: two reviewers are with rating 8. Two reviewers are with rating 4.

Let's examine the two rating 4 reviewers:

Reviewer 4XK6:


1. For the unclear motivation, as I mentioned above, this shall be a misunderstanding and should be addressed.

2. Incomplete description of Benchmark Construction and Data Distribution: The author addressed that in the rebuttal.

3. Practical Deployment considerations: Frankly speaking, I didn't fully understand the statement 'For tokenizer mismatches, token embeddings are aligned by using the base model’s embedding when available; tokens unique to a single model use that model’s embedding; tokens present in multiple models use the averaged embedding.' at first glance. But after re-reading it, I guess what the author tries to say is The embedding contains two parts: the shared parts across models, and the unique parts that belongs to some specific models. For merging the tokenzier, the author chooses to concatenated the average of the shared parts' embeddings and the unique parts embeddings. This should address the reviewer's concern.

Reviewer Kqrb:

1. Innovation: the author addressed it during the rebuttal.

2. The "no data" expression is somewhat exaggerated: the author provided the 'optimal' hyperparameters. I think the author might need to tone down the expression. But this concern should be addressed.

Given that, I think the reviewers should raise their ratings.

---

### Decision · Program_Chairs · 2026-01-26

Accept (Poster)